# Global assessment and mapping of ecological vulnerability to wildfires

Fátima Arrogante-Funes [1], Inmaculada Aguado [1] and Emilio Chuvieco [1]

[1] Environmental Remote Sensing Research Group, Department of Geography and Geology, Universidad de Alcalá, Colegios 2, 28801 Alcalá de Henares, Spain.

*Correspondence to:* Fátima Arrogante-Funes (fatima.arrogante@uah.es)

**Abstract.** Fire is a natural phenomenon that has played a critical role in transforming the environment and maintaining biodiversity at a global scale. However, the plants in some habitats have not developed strategies for recovery from fire or have not adapted to the changes taking place in their fire regimes. Maps showing ecological vulnerability to fires could contribute to environmental management policies in the face of global change scenarios. The main objective of this study is to assess and map ecological vulnerability to fires on a global scale. To this end, we created ecological value and post-fire regeneration delay indices on the basis of existing global databases. Two ecological value indices were identified: biological distinction and conservation status. For the post-fire regeneration delay index, various factors were taken into account, including the type of fire regime, the increase in the frequency and intensity of forest fires and the potential soil erosion they can cause. These indices were combined by means of a qualitative cross-tabulation to create a new index evaluating ecological vulnerability to fire. The results showed that global ecological value could be reduced by as much as 50% due to fire perturbation of poorly adapted ecosystems. The terrestrial biomes most affected are the tropical and subtropical moist broadleaf forest; tundra; mangroves; tropical and subtropical coniferous forests; and tropical and subtropical dry broadleaf forests.

## 1 Introduction

Fire is a natural phenomenon that has played an important role in the transformation of the environment and the maintenance of biodiversity on a global scale. It can have numerous positive and negative impacts. Most of the world's terrestrial habitats where fires occur depend on them for ecological sustainability. (Kirkman et al., 2001; Midgley & Bond, 2015). Fire can affect the distribution of habitats, carbon and nutrient fluxes, and the water-holding properties of soils (Bowman et al., 2009). In habitats that are adapted to and even dependent on fire exclusion policies, this can result in a decrease in biodiversity (Guyette et al., 2002). In addition, the absence of fire results in increases in fuel loads (Bond et al., 2005), which frequently augment the risk of catastrophic fires over time. Fire has also been and continues to be used by humans as a crucial tool for managing terrestrial ecosystems, producing cultural landscapes that also benefit ecological health (Caprio & Graber, 2000; Guyette et al., 2002).

On the other hand, there are some habitats, such as moist tropical forests, that have never adapted to fires. The introduction of fire by humans can lead to an irreparable loss of their structure and composition (Cochrane & Laurance, 2002). Even in fire-adapted areas such as the Mediterranean ecosystems, recent human and climate-related changes in fire regimes are having negative impacts on the functioning of ecosystems (Bajocco et al., 2011; Midgley & Bond, 2015). The increasing frequency and intensity of fires can have negative impacts on forest stands

and landscapes, human life, infrastructures and ecosystem services and wildlife; and can cause changes in regeneration dynamics, hydrological regimes and air quality, among other environmental consequences on a global scale (Alcasena et al., 2016; Barrio et al., 2011; Buhk et al., 2007; Díaz-Delgado et al., 2002; Flannigan et al., 2009; Hobson & Schieck, 1999; Moreira et al., 2011; Scott & Van Wyk, 1990). As a result of this process of change, wildfires have become one of the main environmental problems today at both global and local levels.

This means that fires must be included in global and regional assessments of vulnerability to global change (Houghton et al., 2001; Lindner et al., 2010). Furthermore, fire risk assessment should be carried out spatially in order to design and implement prevention strategies that enable the conservation of the ecological value of ecosystems and landscapes. When fires happen, assessments of this kind can also be useful for implementing post-fire strategies to bring about the recovery of pre-fire ecological values and cultural and socioeconomic assets (Aretano et al., 2015; Chuvieco et al., 2010). In terms of natural hazards terminology, spatially measured fire risk is a combination of 'danger' and 'vulnerability'. 'Danger' is defined as the probability of fire occurring in a given place and time, while vulnerability refers to the potential damage that fire could cause to this place (Chuvieco et al., 2007).

The concept of vulnerability has been studied and applied at different spatial scales and in a wide range of disciplines, in both social and natural studies (Abson et al., 2012; Berry et al., 2006; Cinner et al., 2012; Cutter et al., 2003; Moreno & Becken, 2009).

Vulnerability has many different definitions. For example, the definition proposed by the UNISDIR, (2009)is based on the assumption that an ecosystem cannot cope with a disturbing event (earthquake, fire, flood, etc.) and is therefore vulnerable to it. In order to assess where adaptation actions may be necessary and beneficial, vulnerability assessment must analyse the factors that determine the potential for damage from exogenous threats, as well as the endogenous adaptive capacity of the ecosystem (Preston et al., 2011).

An ideal assessment of ecological vulnerability must therefore take into account the biotic and abiotic aspects of the environment (e.g. species richness, conservation status of the ecosystems), the relationship between them (e.g. ecosystem functionality) (Ippolito et al., 2010), as well as any temporal and spatial pressures (e.g. landscape fragmentation) (Williams & Kapustka, 2000). An integrated approach to vulnerability can therefore be achieved by developing different indices that characterize the biodiversity and ecological quality of the environment and its capacity to adapt and regenerate once a fire has been extinguished.

The integration and harmonization of spatial data of different origin and typology on a global scale in an index is a challenge. Numerous integration techniques exist, such as multicriteria methods (El Gibari et al., 2019). But for a global scale, the lack of a panel that is sufficiently representative of the world would lead to a biased result (depending on the territory of which there was representation or not) (Borrero & Henao, 2017; Hämäläinen & Alaja, 2008). For this reason, qualitative cross-tabulation seems like an integration tool that could be objective enough when dealing with categorical data as proposed by numerous studies (Arrogante-Funes et al., 2021; Martínez Vega et al., 2007).

Some attempts to assess vulnerability do not take all these elements into account (Turner et al., 2003). The study by Duguy et al., (2012) characterized ecological vulnerability using the species richness measurement, at a local scale, in Mediterranean forests. In research in southern Italy, also on a local scale, Aretano et al., (2015) proposed an ecological sensitivity index covering unique habitats, susceptibility to fire and regeneration capacity, but did not evaluate soil erosion after disturbance. At the regional level, Chuvieco et al., (2010) studied ecological

vulnerability in line with the degree of protection of the area, reviewing the different legal forms for the official protection of ecosystems, homogeneous landscape units and land uses. This approach focused more on landscape ecology than on species biodiversity, in which adaptation to fire is considered through the strategies developed by plants in response to fire through the dynamic global vegetation model called ORCHIDEE developed by Krinner et al., (2005). In other research, such as the study by González, Kolehmainen, & Pukkala, (2007), the vulnerability of the ecosystem to fire was evaluated by a group of experts who were provided with images and data on forest metrics measured in the field, together with aerial photographs. Regional studies have been conducted to evaluate the effects of fire on soils and post-fire dynamics in ecosystems (Duguy & Vallejo, 2008; Giovannini & Lucchesi, 1997). The first global analysis of wildfire vulnerability was done by Chuvieco et al. (2014), who estimated the standing ecological value of ecosystems from biodiversity data, their state of conservation and the fragmentation of the landscape. The delay in the post-fire regeneration of vegetation was estimated by assessing their adaptation to fire and potential soil erosion. Adaptation to fire was analysed by comparing the real land cover with fire simulations based on the dynamic global vegetation model.

In this paper, we carry out a systematic assessment of ecological vulnerability to wildfires on a global scale using an index that combines the two main components of vulnerability, namely the ecological value of ecosystems and the delay in post-fire regeneration. The novelty of this approach lies in the characterization of structural biodiversity from the point of view of its exceptionality, while also assessing biodiversity in terms of ecosystem functionality. In addition, in this study, rather than approaching the post-fire regeneration of forests as part of a static, immutable system, as most previous researchers have done, we view these strategies within the dynamic context of changing fire regimes. This study will be carried out on a global scale so as to enable us to tackle the planetary ecosystem as a whole, unrestricted by governmental or geographic borders. In this way, this research could become an essential tool for decision-making about resource management and nature conservation across the globe.

## 2 Materials and methods

### 2.1 Conceptual framework

In order to develop the Ecological Vulnerability Index proposed in this study, our first task was to estimate the ecological value of the environment and its regeneration capacity after fire disturbance. To do so, we had to process the different input variables and devise a way to integrate them into the index (Table 1). In addition, the basic integration tool in the different indicators and index is the qualitative cross tabulation used in many spatial studies (Arrogante-Funes et al., 2021; Martínez Vega et al., 2007).

Table 1: Conceptual framework and diagram for the Ecological Vulnerability Index, and reference sources used in the input variables.

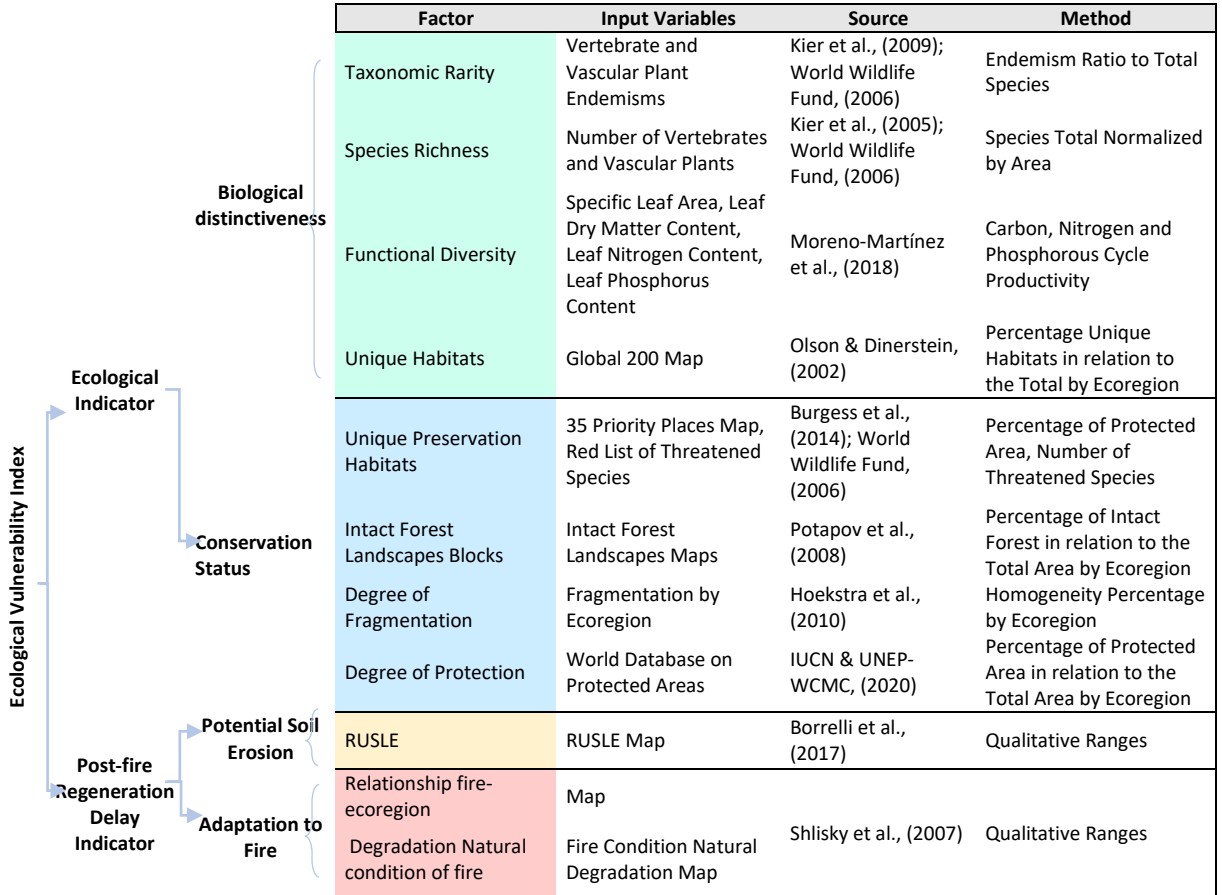

| Factor | Input Variables | Source | Method |
|---|---|---|---|
| Taxonomic Rarity | Vertebrate and Vascular Plant Endemisms | Kier et al., (2009); World Wildlife Fund, (2006) | Endemism Ratio to Total Species |
| Species Richness | Number of Vertebrates and Vascular Plants | Kier et al., (2005); World Wildlife Fund, (2006) | Species Total Normalized by Area |
| Functional Diversity | Specific Leaf Area, Leaf Dry Matter Content, Leaf Nitrogen Content, Leaf Phosphorus Content | Moreno-Martínez et al., (2018) | Carbon, Nitrogen and Phosphorous Cycle Productivity |
| Unique Habitats | Global 200 Map | Olson & Dinerstein, (2002) | Percentage Unique Habitats in relation to the Total by Ecoregion |
| Unique Preservation Habitats | 35 Priority Places Map, Red List of Threatened Species | Burgess et al., (2014); World Wildlife Fund, (2006) | Percentage of Protected Area, Number of Threatened Species |
| Intact Forest Landscapes Blocks | Intact Forest Landscapes Maps | Potapov et al., (2008) | Percentage of Intact Forest in relation to the Total Area by Ecoregion |
| Degree of Fragmentation | Fragmentation by Ecoregion | Hoekstra et al., (2010) | Homogeneity Percentage by Ecoregion |
| Degree of Protection | World Database on Protected Areas | IUCN & UNEP-WCMC, (2020) | Percentage of Protected Area in relation to the Total Area by Ecoregion |
| RUSLE | RUSLE Map | Borrelli et al., (2017) | Qualitative Ranges |
| Relationship fire-ecoregion | Map | Shlisky et al., (2007) | Qualitative Ranges |
| Degradation Natural condition of fire | Fire Condition Natural Degradation Map | | |

## 2.2 Spatial Unit

The spatial units used in this study were the terrestrial ecoregions proposed by the World Wildlife Fund (WWF), as corrected in 2017 (Olson et al., 2001). The terrestrial ecoregion concept refers to a land unit large enough to house a set of natural communities composed of different species, dynamics and similar environmental conditions. Thus, ecoregions are a good way to structure ecological and fire information on a global scale, since they are relatively homogeneous in terms of climate and vegetation (Pausas & Ribeiro, 2017). For this reason, ecoregions are considered a more suitable unit of reference on which to add spatial biological information, compared to other possible units such as grids.

The database is made up of 827 ecoregions distributed in 14 biomes. The ecoregions in which it is impossible for forest fires to occur were excluded, while other areas, such as Antarctica, were excluded due to lack of data. In this way, the final number of ecoregions was 660, having representation of all terrestrial biomes.

## 2.3 Burnable Area

It was necessary to define the burnable area in order to identify areas in which fires are unable to expand. Our assessment of Burnable Area was based on the global Land Cover (LC) dataset produced under the Climate Change Initiative (CCI) program of the European Space Agency (ESA) (www.esa-landcover-cci.org). The CCI-LC map was generated from MERIS-Envisat images acquired at 300 m between 2008 and 2012. The original product

includes 22 land covers, which were reclassified to burnable/unburnable covers and then resampled at a resolution
of 0.25 degrees.
Ecoregions with burnable areas of ≤ 33% were removed from further analysis, as they would suffer only marginal
impacts of fire. This reduced the final number of ecoregions and terrestrial biomes used in our analysis to 647 and
14, respectively (Fig. A1).
**2.4 Representativeness Criteria**
The approach used to establish the ecological value of the different terrestrial ecoregions is based on the concept
of representativeness. In this way, each biome is guaranteed to have at least one priority ecoregion, so ensuring,
for example, that the ecoregions in the savanna forest biome can also be classified, in addition to the more
important moist tropical forests, which would otherwise dominate the list of values due to their high rates of species
richness and endemic species (endemisms). This approach is used in ecoregional evaluations that enable
comparison between studies (Burgess et al., 2006; Ricketts et al., 1999). The biological values were studied by
ecoregion within the biome to which they belong. Then, all the ecoregions with their respective biological values
were combined in a map at global level.
**2.5 Ecological Indicator**
To evaluate the ecological component relative to the ecoregions within each biome, two indicators were
qualitatively generated and integrated by cross-tabulation: i) Biological distinctiveness and ii) Conservation Status.
This approach enables us to characterize structural biodiversity from the point of view of its exceptionality, and in
terms of ecosystem functionality (Dinerstein et al., 1995; Ricketts et al., 1999).
**2.5.1 Biological distinctiveness**
Biological distinctiveness is more than just biodiversity at the species level, in that it also covers the diversity of
ecological functions and the processes that support structural biodiversity (Ricketts et al., 1999). Specifically, this
study is based on taxonomic rarity, species richness, functional diversity, and habitats with a unique evolution.
**Taxonomic Rarity and Species Richness.** The lists of species and endemisms (i.e. at least 75% of the taxon
occurs in the same place) by ecoregion for mammals, birds, reptiles and amphibians form a dataset that can be
gleaned from the literature, distribution databases, and fieldwork carried out by expert taxonomists (WWF, 2006).
Likewise, the data relating to diversity and vascular plant endemisms (Kier et al., 2005, 2009) have been used in
numerous ecological studies (Freudenberger et al., 2012; Poos, Walker, & Jackson, 2009).
To find out more about vertebrate species diversity, the total number was obtained by adding up all the vertebrate
species belonging to the same ecoregion. The data were then normalized according to land area (Eq. (1)):
$$SA = S/(A)^Z \qquad \textbf{(1)}$$
where SA is the number of species corrected by ecoregion, S the total number of species, A is the area in $km^2$ and
Z is the correction factor for continental mainland (value of 0.2) and islands (value 0.25) (Rosenzweig, 1995). As
numerous studies show (Burgess et al., 2006; Olson et al., 2001; Ricketts et al., 1999), the behaviour of this data
type is associated with the size of the territory, which is why in order to make them comparable we had to apply
this method of approximation to the species-area distribution curve. The same process was followed to obtain the
richness of vascular plant species, except that the data for the total number of species by ecoregion had already
been collected.
To assess the absolute taxonomic rarity for vertebrates and vascular plants, the endemism-richness ratio (Eq. (2))
was calculated. This estimates the number of species endemic to the ecoregion as a proportion of its species
richness:

$$R = (\Sigma E / \Sigma S) \quad \textbf{(2)}$$

where R is the percentage of endemisms, E the endemisms and S the species.
**Functional Diversity.** The continuous data about Specific Leaf Area (SLA), Leaf Dry Matter Content (LDMC),
Leaf Nitrogen Content (LNC) and Leaf Phosphorus Content (LPC) (g x g -1) was provided by Moreno-Martínez
et al., (2018) at 500m spatial resolution. It was used as a proxy of Carbon, Nitrogen and Phosphorus cycle
productivity.
To obtain the productivity of each cycle, an average figure by ecoregion was estimated. The productivity values
were then scaled in a monotonous linear manner increasing from 1 to 100, so as to enable us to compare
productivity between the different ecoregions. Finally, functional diversity was integrated as a sum of the
productivity values for the carbon, nitrogen and phosphorus cycle.
The environment is a holistic system, which means that loss of function  affects the capacity of the ecosystem to
support not only itself, but also its neighbours (Pausas & Ribeiro, 2017).  Ecoregions with high functional diversity
values are therefore considered more vulnerable to fires because they provide support for other ecosystems that
could also be damaged indirectly by fire in this way.
**Unique Habitats.** The Global 200 (G200) cartography (Olson & Dinerstein, 2002) shows the area in square
kilometres of habitats with unusual ecological and evolutionary phenomena by ecoregion, which make them
irreplaceable (Myers et al., 2010). In this way, 141 terrestrial ecoregions were identified. To assess the G200
cartography, we calculated the ratio between the area occupied by these habitats and the total area of the ecoregion.
**Integrating the Factors into the Biological Distinctiveness.** The above factors were integrated into the
Biological Distinctiveness  using the criteria established by Burgess et al., (2006). First, the factors were scaled
between 1 and 100 through a linear function per biome. The taxonomic rarity scores were given the most weight
as they establish the qualitative ranges of the biodiversity through quartiles: Very High, High, Moderate and Low
(Table 2). In the case of endemic species, this is because if a fire occurred in one of these ecoregions, the entire
species would be wiped out. For the other factors, the first quartiles of species richness and of unique habitats and
scores of > 95 for functional diversity are taken into account when assigning these ecoregions to the exceptional
category (Table 2).








**Table 2: Summary of the criteria for assigning ecoregions within the biomes to the different categories.**

| Categories | Endemisms | Species Richness | Functional Diversity | Unique Habitats |
|---|---|---|---|---|
| Very High | First quartile of total endemisms within the biome | First quartile of species richness within the biome | Ecoregions with more than 95% productivity | First quartile of unique habitats |
| High | Second quartile of total endemisms within the biome | | | |
| Moderate | Third quartile of total endemisms within the biome | | | |
| Low | Fourth quartile of total endemisms within the biome | | | |

### 2.5.2 Conservation Status

The Conservation Status seeks to estimate the current and future capacity of an ecoregion to meet the following biodiversity conservation and quality objectives: maintain populations and communities of viable species, maintain ecological processes, and respond effectively to environmental changes over time. Specifically, this study is based on the preservation of unique habitats, the presence of landscapes that contain intact habitats, the degree of environmental fragmentation and the level of protection they enjoy.

**Unique Habitats Preservation.** The 35 Priority Places (35PP) cartography, proposed by the WWF, consolidates special conservation areas because they are an extensive and intact representation of unique ecosystems (Burgess et al., 2014). Of these, we maintained the 33 terrestrial ecoregions with a degree of protection and then estimated the ratio between the area occupied by these protected ecosystems and the total area of the ecoregion to which they belonged.

For its part, the Red List of Threatened Species (RL) provides data about the current situation of the biodiversity (WWF, 2006). We maintained the species on this list categorized as: "critically endangered", "endangered" and "vulnerable". These categories were selected because there are common criteria for the management and conservation of the habitats that host these species (Hilton-Taylor, 2000; Mace & Lande, 1991). We then calculated the total number of threatened species by ecoregion.

Both processed variables were scaled from 1 to 100 in an increasing monotonic linear manner and were added together to obtain the singular habitats preservation factor.

**Intact Forest Landscapes Blocks.** From an ecological point of view, old-growth forests are of great importance, albeit more structural than functional, in terms of their role in the conservation of most of terrestrial diversity, hosting indigenous populations and contributing enormously to the regulation of the global climate. Outside these blocks, for example in planted forests, characteristics such as the age of the plants or the composition of the stands could not be maintained in such an exceptional way. The Intact Forest Landscapes (IFL) cartography (Potapov et al., 2008) charts the location and extent of the forests and terrestrial ecosystems that remain unaltered by humans, with a minimum mappable unit of 500 km$^2$. The IFL area data was added to the corresponding ecoregions and the area occupied by these forests as a percentage of the total area of the ecoregion was calculated.

**Degree of Fragmentation.** Landscape fragmentation mapping by ecoregion is based on the method proposed by Hoekstra et al., (2010). It shows the degree of fragmentation as a percentage, with the highest percentages

corresponding to highly degraded or heterogeneous landscapes and the lowest to areas that are unfragmented or
homogeneous.
The degree of homogeneity was established by scaling the values for terrestrial ecoregions in a monotonic linear
manner reversing the original scale from 1-100 to 100-1. The more homogeneous compositions have higher
biodiversity ratios (Collinge, 1996), so making them more vulnerable to fire due to the ecological loss that this
would cause (Pausas et al., 2003).
**Degree of Protection.** Protected status, mainly in the form of national parks and reserves, plays an essential role
in conservation. These areas are mapped in the World Database on Protected Areas (WDPA), which was generated
as part of a project developed by the United Nations Environment Program (UNEP) and by the IUCN, administered
by the World Center Conservation Monitoring Committee (WCMC) and UNEP (IUCN & UNEP-WCMC, 2020).
In this study, we only considered the terrestrial protected areas classified under IUCN categories I-IV, as for these
categories there is reliable data, verified on the ground, and they are managed in a similar way, thus enabling us
to assume that they all have the same biodiversity conservation values. The area data for the WDPAs were added
to the corresponding ecoregions and we then calculated the area occupied by WDPAs as a percentage of the total
area of each ecoregion.
**Integrating the factors into the Conservation Status.** The weights (Table 3) for the different factors (i.e. unique
habitats, intact forest landscapes, degree of fragmentation and degree of protection) and the method for integrating
them into the Conservation Status were as proposed by Burgess et al., (2006) and by Ricketts et al., (1999). These
variables were multiplied by their weight (Table 3) and then added together to obtain the Conservation Status. In
this way, the scores that can be obtained by an ecoregion vary between a minimum of 1 and a maximum of 100
(Table 3). The variables awarded the most weight are those that indicate the quality of an ecosystem in terms of
its size and homogeneity. Then, the values were scaled according to this criterion and qualitative ranges were
generated using quartiles such as Pereira et al., (2020)and Xing & Ree (2017), among others (Table 4).
**Table 3: Values assigned on the basis of conservation status obtained from the G200 cartography**

| Factors | Weights |
|---|---|
| Unique Habitats Preservation | 40 |
| Intact Forest Landscapes | 25 |
| Degree of Fragmentation | 20 |
| Degree of Protection | 15 |

**Table 4: Criteria for assigning ecoregions within biomes to the different categories**

| Categories | Conservation Status |
|---|---|
| Very High | First quartile |
| High | Second quartile |
| Moderate | Third quartile |
| Low | Fourth quartile |

**2.5.3 Integrating the Ecological Indicator**
The Biological Distinctiveness and Conservation Status were constructed using a qualitative cross-tabulation that
prioritized the most valuable elements, given that high biodiversity and quality values also imply high ecological
values in the environment (Ricketts et al., 1999) (Table 5).

**Table 5: Criteria for assigning ecoregions within biomes to the different categories in the Ecological Indicator.**

| | | Conservation Status | | | |
|---|---|---|---|---|---|
| | | Very High | High | Moderate | Low |
| **Biological Distinctiveness** | Very High | Very High | Very High | High | Moderate |
| | High | Very High | Very High | High | Moderate |
| | Moderate | High | High | Moderate | Moderate |
| | Low | High | Moderate | Low | Low |

**2.6 Post-Fire Vegetation Regeneration Delay Indicator**

The delay in the regeneration of vegetation after a fire is an indicator of the difficulties faced by the environment when recovering naturally from fire. It depends on the various strategies adopted by forest species that have adapted to fire and also on the physical state of the soil after the fire. This study provides a dynamic approach which includes an assessment of the alteration of the fire regime. Habitats that have not adapted to the change in fire regimes observed in recent decades will also be assessed.

**2.6.1 Adaptation of the Vegetation to Fire Regimes**

We used the two maps provided by Shlisky et al., (2007), which were generated in collaboration with WWF, the Nature Conservancy (TNC), the University of Berkeley and the IUCN. Firstly, in this database, the ecoregions were grouped into relationship between fire and ecoregion characterized by fire behaviour, plant strategies in response to fire, climatic variables and human use of fire as management tool. Secondly, the ecoregions were grouped together on the basis of the alteration of the natural state of fire regimes, measured in terms of frequency, severity, size and seasonality. The first grouping includes fire-dependent, sensitive and independent ecoregions, while the second classifies ecoregions according to intact, altered and highly altered respect the first grouping.

After reviewing the data base, 647 terrestrial ecoregions were maintained (repeated and confusing information was eliminated, as were ecoregions without data, covered with ice or rock). To estimate the adaptation of the ecoregions to fire regimes, the two factors (regimes and their alteration) were integrated through a qualitative cross-tabulation (Table 6).

The lowest values for Adaptation to Fire Regimes were for the independent and sensitive categories, while the highest were for the ecoregions that were well adapted to fire. In ecosystems that are well adapted to fire, it plays a fundamental role in the conservation of biodiversity. However, in poorly adapted ecosystems, fire can cause serious problems in the recovery and conservation of biodiversity because the plants do not have the necessary strategies to cope with and recover from it (Shlisky et al., 2007).

 **Table 6: Criteria for assigning ecoregions to the different categories of adaptation to fire regimes**

|  |  | **Natural Condition Fire** | | |
|  |  | Very Degraded | Degraded | Intact |
| **Relationship between fire and ecoregion** | Independent | Low | Low | Moderate |
|  | Sensitive | Low | Moderate | High |
|  | Dependent | Moderate | High | Very High |

298

### 2.6.2 Soil Erosion Potential

Post-fire soil erosion can reduce the recovery capacity of the vegetation, and consequently of the ecosystem. The expansion capacity of the roots depends on the quality of the soil, in terms for example of its texture. This is why, after a fire, regeneration of the vegetation does not begin instantaneously. The soil must first recover its original structure and composition and this takes time. The Global Soil Erosion map (Borrelli et al., 2017) was developed using the Revised Universal Soil Loss Equation (RUSLE) with a spatial resolution of 250 m.

Potential soil losses were calculated in tons per pixel. The potential soil erosion per ecoregion (tn / ha) was estimated by adding together all the soil losses and then dividing by the total area. The values were then transformed into a categorical variable according to the criterion for soil erosion due to water, proposed by the Food and Agriculture Organization of the United Nations (FAO) (FAO/UNEP/UNESCO, 1979) (Table 7), which is also applicable to fire erosion processes (Chuvieco et al., 2014).

**Table 7: FAO criteria for assigning ecoregions to different categories of potential soil erosion.**

| Categories | Values (tn/ha year) |
| --- | --- |
| Low | 0 – 20 |
| Moderate | 20 – 50 |
| High | 50 – 200 |
| Very High | > 200 |

### 2.6.3 Integrating the Post-Fire Vegetation Regeneration Delay Indicator

The two factors - Adaptation of Vegetation to Fire and Potential Soil Erosion - were combined by qualitative cross-tabulation (prioritizing the most valuable element) to obtain the Post-Fire Regeneration Delay Indicator (Table 8). This is an indicator of the time required for an ecosystem to regenerate naturally, i.e. for it to recover a structure and composition similar to that that existed pre-fire. Therefore, the higher the delay values, the greater the vulnerability to fire. This factor is the opposite of the post-fire regeneration capacity index calculated by other

authors in local studies (Baeza et al., 2007). Post-Fire Regeneration Delay values from High to Very High were
assigned to ecoregions with a Moderate or Low Adaptation to Fire and High Potential Soil Erosion values. The
lowest Regeneration Delay values corresponded to ecoregions that were well adapted to fire and had low soil
erosion potential.
Table 8: Criteria for assigning ecoregions to the different Post-Fire Vegetation Regeneration Delay categories.

| | | Potential Soil Erosion | | | |
|---|---|---|---|---|---|
| | | Low | Moderate | High | Very High |
| **Adaptation of Vegetation to Fire** | Very High | Low | Low | Moderate | High |
| | High | Low | Low | Moderate | High |
| | Moderate | Moderate | Moderate | High | Very High |
| | Low | Moderate | High | Very High | Very High |


### 2.7 Combining the Ecological Indicator and the Post-Fire Vegetation Regeneration Delay Indicator to form the Ecological Vulnerability to Fire Index

Once the different components of our Ecological Vulnerability to Fire Index had been obtained, they were
combined by means of a qualitative cross-tabulation in which the most valuable component was prioritized (Table
9). In other words, the potential ecological losses due to fires were estimated. The lower the Post-Fire Regeneration
Delay values, the lower the impacts of fire.

**Table 9: Criteria for assigning ecoregions to the different Ecological Vulnerability Index categories.**

| | | Post-Fire Vegetation Regeneration Delay Indicator | | | |
|---|---|---|---|---|---|
| | | Low | Moderate | High | Very High |
| **Ecological Indicator** | Low | Low | Low | Moderate | High |
| | Moderate | Low | Moderate | Moderate | High |
| | High | Moderate | High | Very High | Very High |
| | Very High | High | High | Very High | Very High |


2.8 Sensitivity Analyses: One at a time
The objective of a sensitivity analysis is to test the uncertainty of the result of a mathematical model due to the
integration of numerical variables. The one-at-a-time (OAT) method is the most widely used in the literature and
consists of analysing the effect of making small variations on one input while others remain fixed (Saltelli et al.,
339  2000).

In this study, the variables that make up the Ecological Fire Vulnerability Index are of a categorical type and it is
for this reason that a modification of the OAT method is proposed in order to be able to estimate the uncertainty
of the product such as Gonzalez et al., (2015) and Clavijo et al., (2019) estimated in theirs studies. In the way of
integrating said index through the Ecological and Post-fire Regeneration Delay indicators, the resulting label of
ecological vulnerability obtained through the qualitative cross tabulation has been varied (Table 10). In this way
we will be able to establish stable ecoregions (reference) and changing ecoregions (uncertainty).

**Table 10: Criteria for assigning ecoregions to the different Ecological Vulnerability Index categories in**
**order to test the OAT.**

**Post-Fire Vegetation Regeneration Delay Indicator**

|  |  | Low | Moderate | High | Very High |
|---|---|---|---|---|---|
| **Ecological Indicator** | Low | Low | Moderate | Moderate | High |
| | Moderate | Moderate | Moderate | High | High |
| | High | Moderate | High | High | Very High |
| | Very High | High | High | Very High | Very High |


The changes made correspond to: (i) the same category of label corresponds to the same resulting label, (ii) if two
continuous categories face each other, the resulting label will be the one with the highest category and (iii) between
two different categories the label of resulting vulnerability will be an intermediate category, prioritizing the highest
when there are several in between.

## 3 Results

### 3.1 Ecological Indicator

Figure 1 shows the Ecological Value by ecoregion in terms of Biological Distinctiveness (Fig. A2) and
Conservation Status (Fig. A3) indices. Ecoregions of increasing ecological value are shown in a range of tones
from light green to dark green.

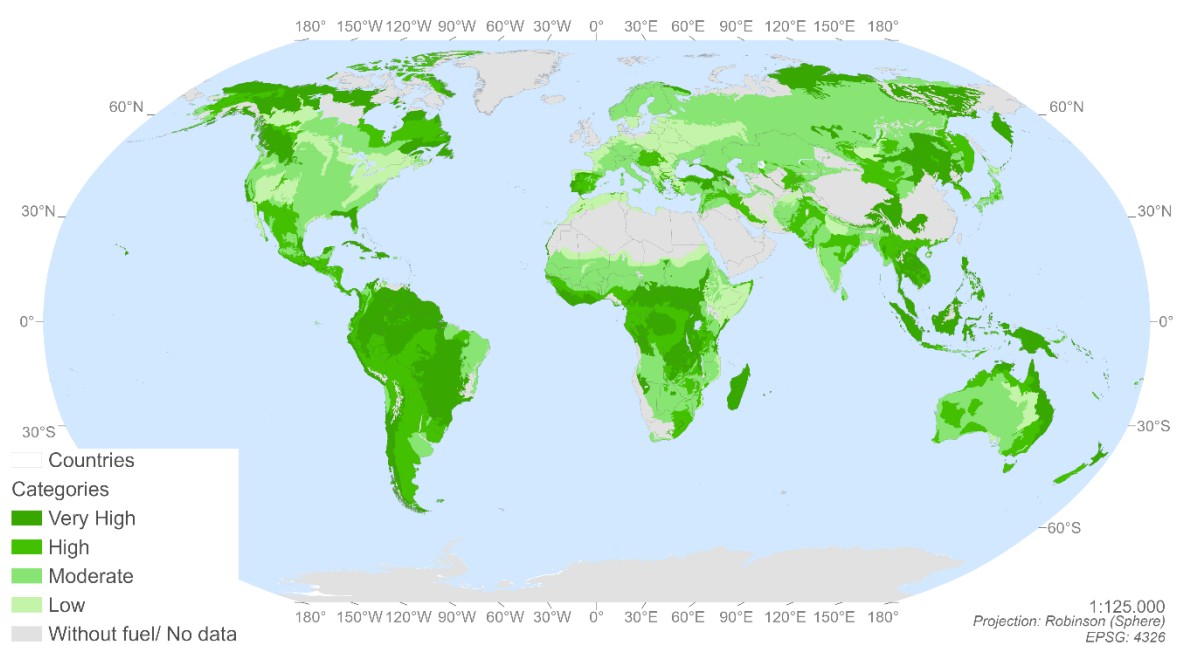

**Figure 1: Spatial distribution of Ecological Value by ecoregion (Ecological Indicator) calculated by combining the Biological Distinctiveness (by ecoregion evaluated within the biome to which it belongs) and the Conservation Status (by ecoregion).**

There are 220 ecoregions with Very High Ecological values, 163 with High values, 206 with Moderate values and 59 with Low values. The ecoregions with the highest Ecological Values (Fig. 1) are located in temperate zones, such as British Columbia, forests in the US and European Mediterranean, China, New Zealand; and in the tropical and subtropical regions, for example the Amazon Basin, Sierra Leone, Cameroon, the Congo Basin, Zambia, Madagascar, New Guinea and northern Australia. Boreal areas, such as Canada and Russia, also show high ecological values.

**3.2 Post-Fire Regeneration Delay Indicator**

Figure 2 shows the Post-Fire Regeneration Delay value by ecoregion, in terms of Adaptation of Vegetation to Fire (produced by combining the plant strategies and fire-regime alteration factors) (Fig. A4) and Potential Soil Erosion (Fig. A5). The very high and high Delay values, highlighted in dark purple tones, are for areas with high Erosion and low Adaptation to Fire, while the moderate and low values, highlighted in lighter lilac tones, are associated with vegetation with very high and high Adaptation to Fire values and moderate or low Erosion values.

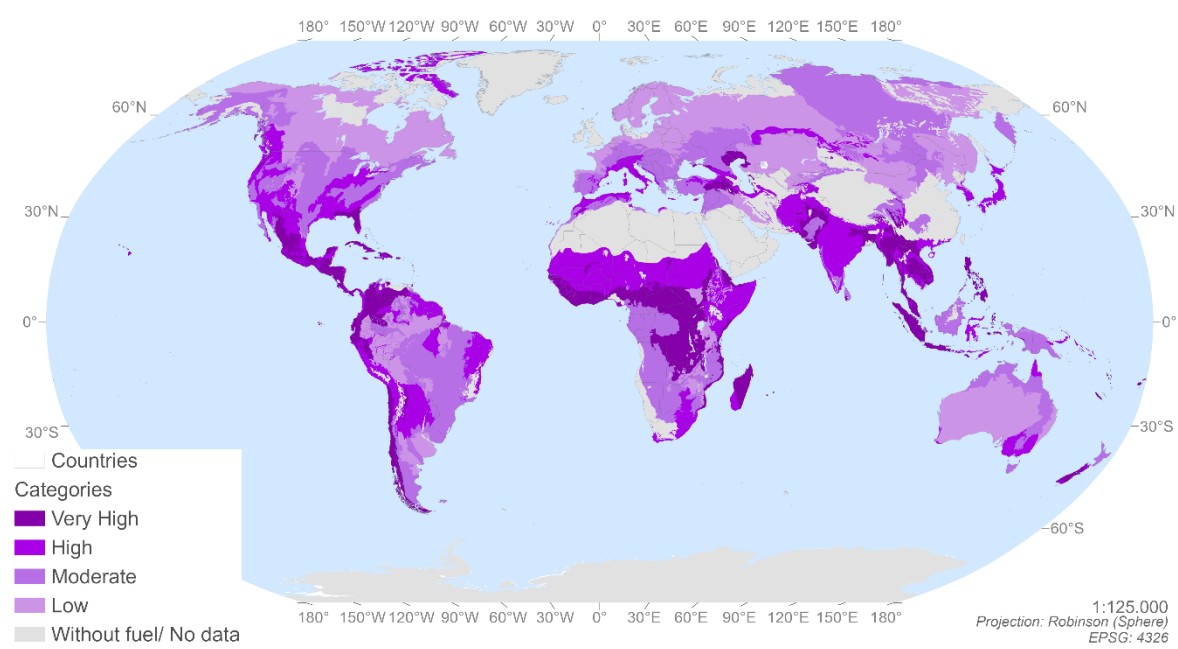

**Figure 2: Spatial distribution of Post-fire Regeneration Delay Values by ecoregion calculated by combining the Adaptation to Fire and the Potential Soil Erosion values by ecoregion.**

Of the 647 ecoregions evaluated, 154 had very high Post-fire Regeneration Delay values, 271 had high values, 157 had moderate values and 120 had low values. The least resilient zones (with low or moderate Adaptation to Fire and high or very high Potential Soil Erosion) belonged to temperate regions such as Florida, the Yucatan Peninsula, eastern United States, the forests of California, Chile and the Spanish Mediterranean and forests in the Caucasus, Himalayas and New Zealand; and in tropical and subtropical areas, for example in Colombia, Ecuador, the Congo Basin, Zambia, Tanzania, Madagascar, countries bordering the Tibet Autonomous Region, the Philippines, Bangladesh, India and New Zealand.

By contrast, the most resilient areas of the planet (very high or high Adaptation to Fire values and low or moderate Potential Soil Erosion) are in the boreal forests of Canada and Russia, Mediterranean forests, the woodlands and scrubs of southern Australia, and the temperate grasslands, savannas and shrublands of Euro-Asia.

**3.3 Ecological Vulnerability to Fire Index**

**3.3.1 Spatial distribution**

Figure 3 shows the Ecological Vulnerability to Fires values by ecoregion (from Ecological Vulnerability to Fire Index). These values were calculated by combining the delay in post-fire regeneration and the ecological indicator values. In other words, this map shows the intensity of potential damage and the capacity to regenerate after wildfires. The areas with the highest values are shown in dark red and correspond to those with significant Post-Fire Regeneration Delay values and high Ecological values. By contrast, the areas shown in lighter salmon tones correspond to ecoregions that are not particularly vulnerable to fire and would incur few potential ecological losses, since they have low Ecological and low Post-Fire Regeneration Delay values.

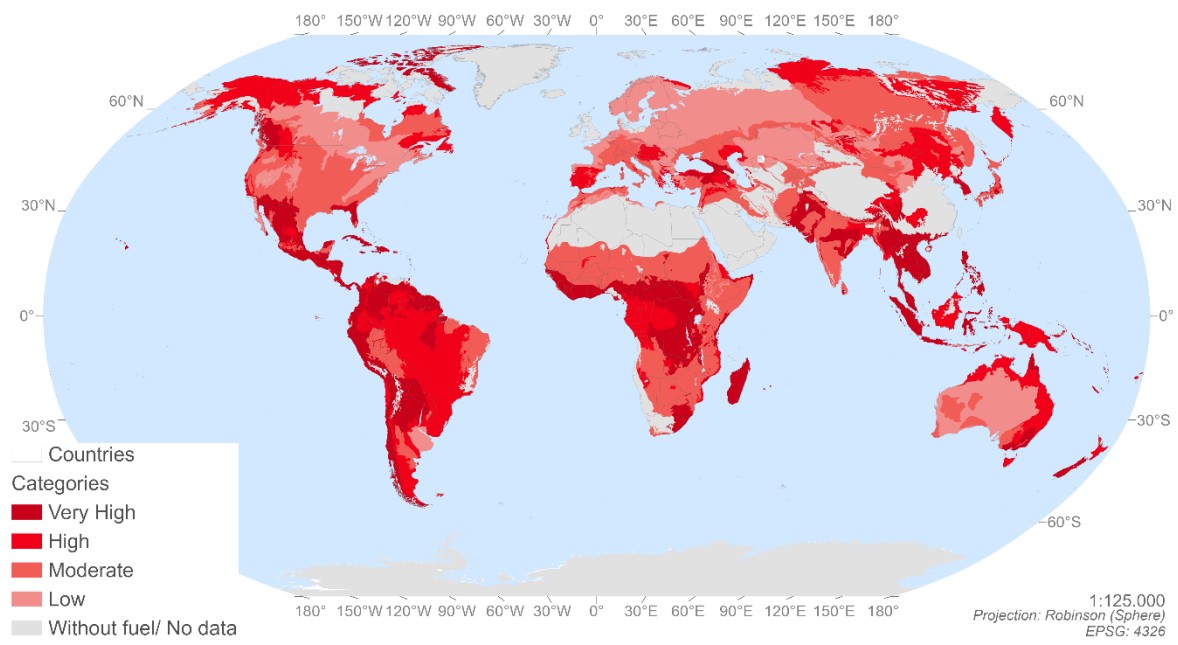

**Figure 3: Spatial distribution of Ecological Vulnerability to Fire Index values calculated by combining the Post-Fire Regeneration Delay and the Ecological Indicators values by ecoregion.**

Of the 647 ecoregions analysed, 246 had very high Vulnerability to Fire values, 155 had high values, 182 were moderately vulnerable and 77 had low values. The areas that would suffer the greatest ecological losses per biome in the event of fire are the temperate zones of British Columbia, the Himalayas, central China, California, Spain, South Africa, Florida, South Sudan, New Zealand, Mongolia, eastern Australia, Chile, Hungary, Romania, Croatia, Serbia, Italy and the Caucasus area; and tropical and subtropical areas such as Mexico, Central America, the Amazon Basin, Sierra Leone, Cameroon, Guinea, the Congo Basin, Paraguay, Argentina, Uruguay, Madagascar, Borneo, Sumatra, the Philippines, Namibia and northern Australia. The ecosystems of the Canadian and Russian boreal forests and the Bolivian and Chinese montane grasslands and shrublands are also vulnerable to fire.

**3.3.2 Biome area assessment**

Almost 50% of the ecoregions have either very high or high Ecological Vulnerability to Fire values (calculated by combining the Post-Fire Regeneration Delay and the Ecological indicators), while only 21% of ecoregions have low Ecological Vulnerability to Fire. This is due to an increase in the frequency and intensity of large Wildfires. The terrestrial biomes that contain most land in the very high and high Vulnerability categories as a proportion of their total area are: tropical and subtropical moist broadleaf forests, tundra, mangroves, tropical and subtropical coniferous forests, and tropical and subtropical dry broadleaf forests.

Within the very high Vulnerability to Fires category, the dominant terrestrial biomes are: tropical and subtropical moist forests, tropical and subtropical grasslands, savannas and shrublands, and xeric shrublands. By contrast, the least common biomes in this category are: wetlands, temperate grasslands, savannas and shrublands, and mangroves. Boreal forests do not have any areas with very high vulnerability values.

Of the 106,605,491 km$^2$ considered in this study (Table 11), the area classified as having very high vulnerability to fires consisted (from highest to lowest) of 7,611,385 km$^2$ of tropical and subtropical moist broadleaf forests,

5,905,304 km$^2$ of tropical and subtropical grasslands, savannas and shrublands, 1,980,099 km$^2$ of xeric shrublands,
1,593,959 km$^2$ of tropical and subtropical dry broadleaf forests, 1,300,302 km$^2$ of temperate broadleaf and mixed
forests, 1,170,778 km$^2$ of temperate conifer forests, 1,053,305 km$^2$ of montane forests and shrublands, 556,032
km$^2$ of tundra, 524,545 km$^2$ of tropical and subtropical conifer forest, 172,422 km$^2$ of Mediterranean forests,
woodlands and scrubs, 154,022 km$^2$ of mangroves, 87,651 km$^2$ of temperate grasslands, savannas and scrublands;
and finally 25,131 km$^2$ of flooded grasslands and savannas.
By contrast, if we look at the different biomes (Table 11), the most vulnerable (from highest to lowest) are as
follows: Tropical and subtropical moist coniferous forests with 75.07% of their area classified as very high
vulnerability, mangroves with 59.61%, tropical and subtropical dry broadleaf forests with 53.08%, tropical and
subtropical moist broadleaf forests with 41.82%, montane grasslands and shrublands with 33.83%, temperate
conifer forests with 29.65%, tropical and subtropical grasslands, savannas and shrublands with 29.27%; xeric
shrublands with 14.02%, tundra with 13.55%, temperate broadleaf and mixed forests with 12.22%, Mediterranean
forests, woodlands and scrubs with 5.38%, flooded grasslands and savannas with 2.93% and, lastly, temperate
grasslands, savannas and shrubs with 0.88%. None of the 'Boreal forests and taigas' biome falls within the very
high vulnerability category, but around 20% of its area is classified as high vulnerability.
As regards the biomes with the lowest Vulnerability to Fire values as a proportion of their total area (Table 11),
the temperate broadleaf and mixed forests stand out (44.85%) followed by boreal forests and taiga (41.37%), xeric
shrublands (35.01%), and Mediterranean forests, woodlands and scrubs (31.85%). The mangroves biome is also
worth highlighting in that its entire area is vulnerable to fire (Table 10).





















**Table 11: Number of ecoregions, surface area and percentage of land ecologically vulnerable to fires.**

| Biome | Percentage of total area studied by biome | Vulnerability Categories | Ecoregions | km² | Percentage per biome |
|---|---|---|---|---|---|
| Tropical & Subtropical Moist Broadleaf Forests | 92.05 | Very High | 105 | 7611385 | 41.82 |
| | | High | 55 | 8318171 | 45.70 |
| | | Moderate | 20 | 1972358 | 10.84 |
| | | Low | 3 | 300554 | 1.65 |
| Tropical & Subtropical Dry Broadleaf Forests | 99.77 | Very High | 28 | 1593959 | 53.08 |
| | | High | 11 | 454328 | 15.13 |
| | | Moderate | 9 | 929016 | 30.94 |
| | | Low | 1 | 25432 | 0.85 |
| Tropical & Subtropical Coniferous Forests | 98.52 | Very High | 12 | 524545 | 75.07 |
| | | Moderate | 2 | 174236 | 24.93 |
| Temperate Broadleaf & Mixed Forests | 82.96 | Very High | 17 | 1300302 | 12.22 |
| | | High | 14 | 1600054 | 15.03 |
| | | Moderate | 19 | 2970276 | 27.91 |
| | | Low | 16 | 4773459 | 44.85 |
| Temperate Conifer Forests | 96.62 | Very High | 19 | 1170778 | 29.65 |
| | | High | 4 | 558328 | 14.14 |
| | | Moderate | 20 | 1369471 | 34.69 |
| | | Low | 6 | 849432 | 21.52 |
| Boreal Forests & Taiga | 94.85 | High | 12 | 2753116 | 19.19 |
| | | Moderate | 5 | 5659834 | 39.45 |
| | | Low | 8 | 5935488 | 41.37 |
| Tropical & Subtropical Grasslands, Savannas & Shrublands | 100.00 | Very High | 14 | 5905304 | 29.27 |
| | | High | 12 | 4217891 | 20.90 |
| | | Moderate | 16 | 9362256 | 46.40 |
| | | Low | 3 | 691856 | 3.43 |
| Temperate Grasslands, Savannas & Shrublands | 98.26 | Very High | 2 | 87651 | 0.88 |
| | | High | 8 | 2631992 | 26.52 |
| | | Moderate | 18 | 4622103 | 46.57 |
| | | Low | 8 | 2584338 | 26.04 |
| Flooded Grasslands & Savannas | 78.70 | Very High | 2 | 25131 | 2.93 |
| | | High | 4 | 425610 | 49.54 |
| | | Moderate | 5 | 250872 | 29.20 |
| | | Low | 3 | 157458 | 18.33 |
| Montane Grasslands & Shrublands | 60.01 | Very High | 16 | 1053305 | 33.83 |
| | | High | 5 | 628994 | 20.20 |
| | | Moderate | 14 | 1089028 | 34.98 |
| | | Low | 2 | 341828 | 10.98 |
| Tundra | 35.20 | Very High | 2 | 556032 | 13.55 |
| | | High | 11 | 2916345 | 71.09 |
| | | Moderate | 3 | 385270 | 9.39 |
| | | Low | 1 | 244865 | 5.97 |
| Mediterranean Forests, Woodlands & Scrubs | 99.47 | Very High | 3 | 172422 | 5.38 |
| | | High | 5 | 624670 | 19.50 |
| | | Moderate | 21 | 1385415 | 43.25 |
| | | Low | 9 | 1020796 | 31.87 |
| Xeric Shrublands | 50.64 | Very High | 13 | 1980099 | 14.02 |
| | | High | 8 | 882566 | 6.25 |
| | | Moderate | 23 | 6314163 | 44.71 |
| | | Low | 14 | 4944312 | 35.01 |
| Mangroves | 74.59 | Very High | 9 | 154022 | 59.61 |
| | | High | 3 | 55773 | 21.58 |
| | | Moderate | 4 | 48602 | 18.81 |
| **Total** | 78.85 | | | 106605491 | |

**3.4 Sensitivity analysis: OAT**

Table 11 shows the results of the sensitivity analysis called OAT carried out through the qualitative cross-tabulation method between Ecological and Post-Fire Regeneration Delay Indicator in order to obtain the Ecological Vulnerability to Fire Index. The categories of the Ecological Vulnerability to Fire Index that present the greatest changes are: High, reaching higher numbers of ecoregions (+95) and Low, decreasing its number of ecoregions considerably to 14 (-65). The number of stable ecoregion per category of Ecological Vulnerability to Fire (obtain the same tag in the Ecological Vulnerability to Fire Index and then, in the OAT sensitivity method) that represent ecoregion of reference are: 185 of Very High, 152, of High, 159 of Moderate and 14 of Low. The total of it reaching 510 ecoregion stables from the 647 ecoregion of this study (Fig. A6). Thus, the percentage of matches is 80.37%.

**Table 11: Accuracy of the model, number of ecoregions per category of Ecological Vulnerability to Fire from the Index and Sensitivity method, and number of stable and net change of ecoregion between the Index and Sensitivity method.**

| Categories of Ecological Vulnerability | Number of ecoregion of the Ecological Vulnerability Index | Number of ecoregion of sensitivity of Ecological Vulnerability Index | Number of stable ecoregion per category of Ecological Vulnerability Index | Net change of ecoregion per category of Ecological Vulnerability Index |
|---|---|---|---|---|
| Very High | 247 | 185 | 185 | -62 |
| High | 194 | 289 | 152 | 95 |
| Moderate | 127 | 159 | 159 | 32 |
| Low | 79 | 14 | 14 | -65 |
| Total of ecoregions | 647 | 647 | 520 | - |
| Matches (%) | | 80,37 | | |

**4 Discussion**

This study presents an index for assessing and mapping Ecological Vulnerability to Fire on a global scale on the basis of Ecological Indicator and Post-Fire Regeneration Delay Indicator. Our results show that global ecological value may be reduced by as much as 50% due to the perturbation by fire of ecosystems that are poorly adapted to fire and have degraded fire regimes. The terrestrial biomes most affected are the tropical and subtropical moist broadleaf forest, tundra, mangroves, tropical and subtropical coniferous forests, and tropical and subtropical dry broadleaf forests. The most important determining factor is fire regime, in that current alterations to the fire regime are causing areas that were previously considered to be relatively safe to now be classified as vulnerable to fire.

This study attempts to evaluate Ecological Vulnerability to Fire on a global scale. Although the databases used were carefully examined before selection, the results are inevitably affected by the different spatial units, the lack of information, the lack of updating and the uncertainty in the data for some ecoregions; and to a lesser extent, by the way we combined the factors in the different indices.

In order to avoid problems with estimations of Species Richness, we used field data which measured this variable exactly. In comparison with the use of remote sensing data, the study by Duro et al., 2007 shows that the Net

Primary Productivity (NPP) value overestimates biodiversity in areas covered by reforestations. This is because
forests made up of young trees or saplings, which fix more carbon than mature forests, are being overestimated.
In addition, the NPP biodiversity values are evaluated in terms of the number of different individuals and not in
terms of the number of different species, a fundamental indicator for establishing the biodiversity values of
particular environments (Nagendra & Rocchini, 2008).
As regards the ecosystem functionality variables, remote sensing data has the advantage of providing updated
information for the entire planet. Despite the extensive bibliographic review carried out as part of this research,
we were unable to find a concise way of combining these variables due to the fact that little research has been done
on the specific issue of ecosystem functionality. This is one of the first studies of ecological vulnerability to fire
that takes this issue into account, by integrating it into ecological value. This is of the utmost importance since fire
affects both the functioning of the ecosystem and its ability to maintain itself (Pausas & Ribeiro, 2017).
Our Ecological Vulnerability to Fire Index highlights those biomes considered most susceptible (tropical and
subtropical moist broadleaf forests, tundra, mangroves, tropical and subtropical coniferous forests, and tropical
and subtropical dry broadleaf forests) to suffering a decline in their ecological value. Two clusters can be observed.
The first consisted of mangroves and tropical and subtropical forests associated with tropical latitudes. These
regions obtained high or very high Ecological Vulnerability to Fire values due to the fact that they had the highest
ecological values and also had high regeneration delay values. Within the ecological value dimension of this index,
tropical latitudes show the highest values for both Biological Distinctiveness and Conservation Status due to the
fact that they host the highest ratios for biodiversity and endemisms, and have high ecosystem functionality values
and low levels of landscape degradation. They also have high levels of official protection. In addition, these areas
have the highest regeneration delay values due to the low adaptation capacity of the vegetation, the high current
alterations of the natural fire regime and the high potential soil erosion after fire disturbance. For this reason, if a
wildfire occurs in biomes such as mangroves, tropical and subtropical moist and broadleaf forests, and coniferous
forests, the ecological value of these biomes will almost certainly be heavily degraded due to the fact that most
areas within these biomes fall within the very high Ecological Vulnerability to Fire category of our index.
Second on this list of the biomes with the largest area with a high potential for degradation by fire is tundra, due
to the fact that it scores highly in both Ecological Value and Regeneration Delay, the two components of our
Vulnerability to Fire index. In terms of the first component, the intrinsic behaviour of the tundra biome explains
why it has similar ecological values to the biomes in the first cluster. However, the high levels of Regeneration
Delay have a different explanation. Within the Regeneration Delay Indicator, tundra has a fire regime in which the
vegetation is well adapted to fire due to the fact that, unlike the tropical and subtropical biomes, frequent fire
disturbance has been a constant feature of its development. In spite of this, tundra biomes have large areas in the
high or very high Vulnerability to Fire categories due to the fact that they score high values for potential soil
erosion and fire regime modification. As a result, pre-fire ecological values will be difficult to recover if the
wildfire occurs under a different regime than that to which the vegetation has adapted. This is why large swathes
of the tundra biome are classified within the very high Vulnerability to Fire category of our index.
In the end, both clusters meet the two requirements of our index for them to be considered highly vulnerable to
losing their pre-fire ecological values in the event of perturbation by fire: (i) high Ecological Indicator values and
(ii) high Regeneration Delay values. Within the Ecological Indicator, the factors which led the different ecoregions
to obtain high Ecological Indicator values are related to the ability of their ecosystems to host different kinds of

plants and wildlife (endemisms, functional and structural biodiversity) and the degree of official protection afforded to them. For its part, the factor with the greatest impact on Regeneration Delay values is the alteration of the fire regime, as this means that the strategies developed by the vegetation in response to fire are no longer fit for purpose, and cannot help it recover the Ecological Indicator values existing prior to the fire. This is why alteration of the fire regime is the most important factor and the most closely associated with human action in that it is largely a consequence of human-induced global change. In this context, a determined shift towards more sustainable lifestyles would reduce ecological vulnerability to fire.

In this sense, up to 50% of the terrestrial ecosystem analysed in this study is vulnerable to potential degradation of its ecological value due to the changes taking place in fire regimes. This estimate coincides with the climate change projections that indicate an increase in the frequency and intensity of large forest fires, recently dubbed "megafires", as a result of longer, drier fire seasons (Stephens et al., 2013, Aponte et al., 2016). This increase, at least in the medium term, will lead to new fire regimes and an increase in aridity in some regions as a consequence of climate change (Flannigan et al., 2009). Terrestrial ecosystems will need to adapt not only to changes in mean climatic variables, but also to greater variability with increased risk of extreme weather events, such as prolonged droughts, storms, and floods (Lindner et al., 2010). As a result of this process of change, forest fires have become one of the main environmental problems at a global scale today.

If we compare our evaluation of Ecological Vulnerability to Fire Index with the study carried out by Chuvieco et al., (2014), substantial differences can be observed. Firstly, in our study the temperate conifer forests in the British Columbia region had high vulnerability values compared to those estimated with their index. Lightning fires are frequent in this area and the ecosystem has learnt to adapt to them. However, in our study, we included the possibility of change in the fire regime, which indicates that these areas are in fact quite vulnerable to fire. Nitschke & Innes (2013) found that due to climate change, fire regimes in boreal areas are changing in frequency and area. If we look for example at the temperate broadleaf and mixed forests of Patagonia and the boreal forests of Alaska, although both have adapted to fire to some extent, they also obtained high vulnerability to fire values, because of the alteration in their fire regimes due to climate change, as indicated by Higuera et al., (2009) and Landesmann et al., (2015).

If we turn our attention to the tropical and subtropical dry broadleaf forests of India, one of the greatest biodiversity areas in the world, in the study by Chuvieco et al., (2014) they were considered to have low vulnerability to fire because their plant communities had adapted to it. However, our study offers a different assessment, awarding these parts of India higher Ecological Vulnerability to Fire values. This may be due to the fact that our model takes into account a variable that characterizes the delay in post-fire regeneration as a result of changes in the fire regime. In this sense, Kodandapani, Cochrane, & Sukumar (2008) refer to the fact that logging and forest fragmentation, grazing and the collection of non-timber forest products are affecting the behaviour of fire in these forests.

In relation to the Amazon Basin, in this study the highest vulnerability to fire values only occur in the regions close to the mouth. This may be due to the way in which the Species Richness variable is characterized. Species Richness, adjusted in line with the size of the ecoregion, enables us to compare ecoregions of different sizes so as to assess the ecological value fairly, rather than just comparing the raw data (Ricketts et al., 1999). It should be noted that the areas near the coast, which have a more open plant canopy that allows sunlight to penetrate, have managed to develop undergrowth vegetation that supports other forms of life (greater species richness understood as diversity of species rather than abundance of species). In this case, it is important to realize that we are dealing

with tropical and subtropical moist broadleaf forests, which have not developed in the presence of fire. The introduction of fire into these ecosystems could therefore result in significant losses in that plant species have never developed post-fire regeneration strategies. This is why the small ecoregions at the mouth of the Amazon suffer slightly greater losses due to fire, compared with the large central ecoregions (Cochrane & Laurance, 2002). In addition, in the present study, the large temperate broadleaf and boreal forests of northern Europe and Russia show less ecological vulnerability to fires than estimated by Chuvieco et al. (2014). This may be due to the fact that our model, by following a representative criterion of estimating the ecological value within the biome, gives higher species ratios to smaller regions, and less weight to the large ecoregions in northern Europe and Russia. This is why, in our study, on a global scale, these ecoregions obtained a low vulnerability to fire value given that to destroy all their ecological wealth, their entire immense area would have to be affected.

As for our index, despite the similarities and differences in the results with other studies, it has its own uncertainty like all models. From the sensitivity analysis, it could be said that approximately 80% of the ecoregions evaluated with the Ecological Vulnerability to Fire index would be considered robust. On the other hand, of the small changes made, around 20% of the ecoregions would show uncertainty in the result of the index.

For example, some of them are located in Africa. Focusing on them, it is surprising to see Zambia and NE Angola mapped with a very high Post-fire Regeneration Delay, especially considering how often they burn. Another example would be that the most resilient areas on the planet (very high or high Fire Adaptation values and low or moderate Potential Soil Erosion) are found in the temperate broadleaf and mixed forests of northern Europe when fire is a rare event in these ecoregions and thus lack a history of fire-attuned evolution. Given the global scale, the heterogeneity of the sources used and the extensive area that an ecoregion represents, sometimes the uncertainty does not come from the integration method but from the prior uncertainty of the databases to be used (Richards & Rowe, 1999). On the other hand, it should be noted that the use of the global scale gives us general information on what is happening in order to detect points of controversy on which to proceed to a study at a local/regional scale (Goodchild et al., 1993). Despite this, these uncertainties will be explored for future versions.

All integration methods, both quantitative and in our case qualitative (cross tabulation), show uncertainty in their results, but as the literature points out, it is necessary to deal with it (Heuvelink, 1998; Heuvelink et al., 1989).

At various points in our study, we combined different factors to create an index. Although the model is based on the bibliography, improvements such as multi-criteria evaluations involving expert participation could be applied in the future in a bid to enrich the proposed approach in local/regional scales (Gómez-Delgado & Tarantola, 2006). We could also apply machine-learning techniques to enable us to establish a more precise relationship between the different factors (Semeraro et al., 2016). For all of the above, the resulting estimates should be interpreted as an initial approximation.

Despite the aforementioned limitations, this study presents a robust, pragmatic and easily understood aggregation methodology. The negative effects of fires can only be identified after the fire. This means that a model of ecological vulnerability to fire cannot be correctly validated on a global scale as there is no representative sample for doing so. However, at regional and local scales, there are studies that monitor post-fire ecological damage (Gouveia et al., 2010). This is because the effects of fire can best be understood at these scales. As this methodology can be replicated easily and the factors can be adapted to the model (to a greater or lesser extent depending on the information available), the model could and indeed should be validated at these scales.

The ecological vulnerability model at a global scale is also very useful as it can help us to understand the global
impacts that fires could have on ecosystems and on climate change. In addition, on a global scale, there are studies
that focus on the early detection of places where fires may occur (based on climate data) (De Groot et al., 2006).
If these studies were combined with our map, they could help prevent or mitigate ecological losses, as well as
encourage the development of action plans in the event of fire, aimed at accelerating the regeneration of the
ecosystem.
This model could also be used in the field of forest management to prioritize fire intervention areas in terms of
ecological value, as proposed by Burgess et al. (2006) and Ricketts et al. (1999). If this vulnerability index were
included in fire management plans, in the event of several fires breaking out at the same time, priority action could
be directed at the most vulnerable area in order to protect its ecological value. Although in these cases, the
protection of human lives is normally the first priority, future studies are expected to develop and integrate the
idea of socioeconomic vulnerability into this ecological component of vulnerability. It would therefore seem more
logical to develop policies, prevention and restoration plans in the most vulnerable areas in order to preserve them.
Although this model for evaluating ecological vulnerability to fires on a global scale is an initial approximation, it
allows us to identify which ecoregions of the different biomes are more likely to have their ecological value
impaired by fire and why.
**5 Conclusions**
This paper makes an initial assessment of the spatial distribution of ecological vulnerability to fire on a global
scale. The methodology we implemented enabled us to systematically integrate all the ecological components
likely to be affected by forest fires. A novel aspect of this methodology is the way it integrates the variables in the
biological distinction index, the characterization of functional diversity and the fact that it takes into account the
impact of the alteration of the natural condition of the fire in post-fire regeneration delay. This index made it
possible to identify the most susceptible biomes in terms of the loss of their ecological values, and it could be
useful as a starting point for developing plans and strategies in response to global change scenarios.
At a global level, our results show that almost 50% of the world's land surface is vulnerable to a decline in its
ecological value due to fire as a result of the current alteration of the fire regime. The terrestrial biomes with the
highest degree of ecological vulnerability to fire were found in the tropical and subtropical moist broadleaf forests;
tundra; mangroves; tropical and subtropical coniferous forests; and tropical and subtropical dry broadleaf forests.
The greatest determining factor is the fire regime, a problem that is being exacerbated by current alterations, in
that areas that were previously considered to be relatively safe now have much higher vulnerability values due to
alterations in the natural condition of the fire, caused by global climate change.

 **Appendix A: Maps of the study area, indicators and sensitivity method**

In this section, we show the maps produced by the study area, indicators and sensitivity method (Fig A1-6).

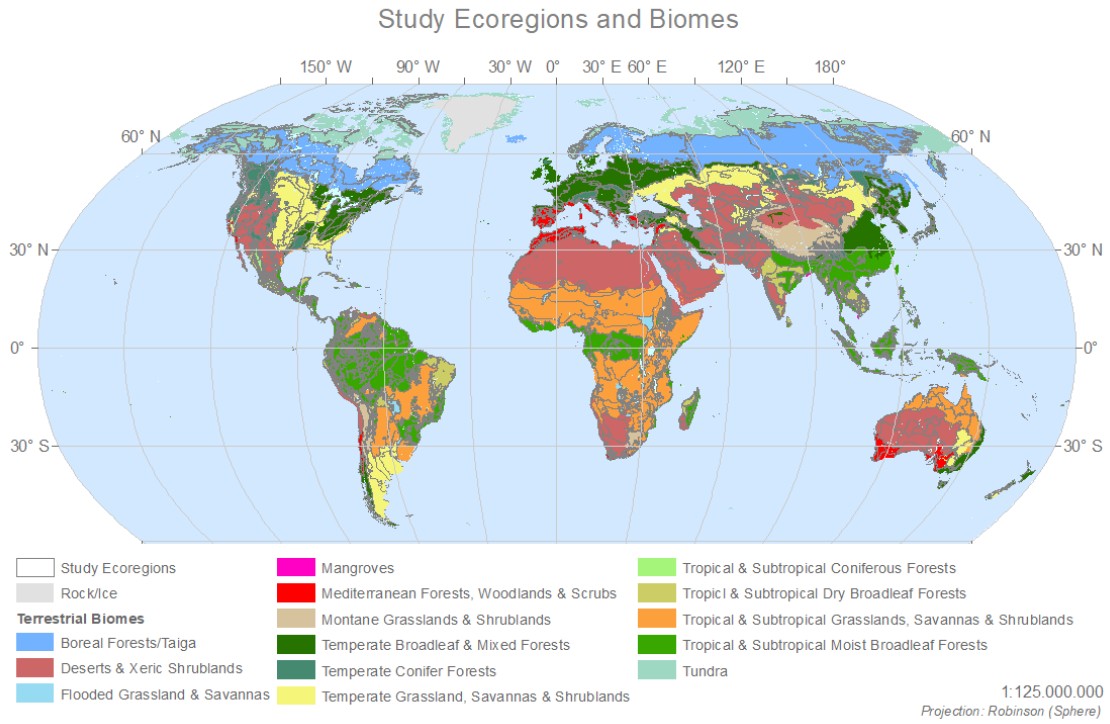


**Fig. A1: Terrestrial ecoregions within their respective biomes for this study. (Source: The authors).**

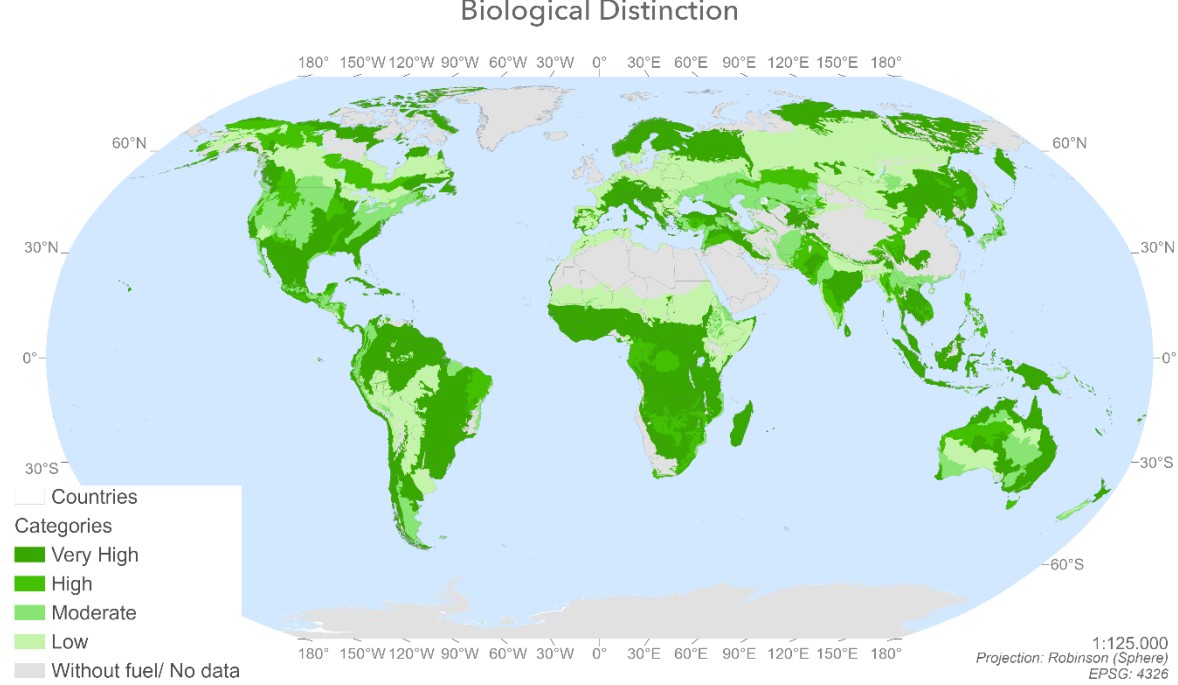


**Fig. A2: Spatial distribution by ecoregion of the Ecosystem Biological Distinction Value prepared by**
**combining the indices for Endemic Species, Species Richness, Functional Diversity and Unique Habitats by**
**ecoregion evaluated within the biome to which they belong. (Source: The authors).**

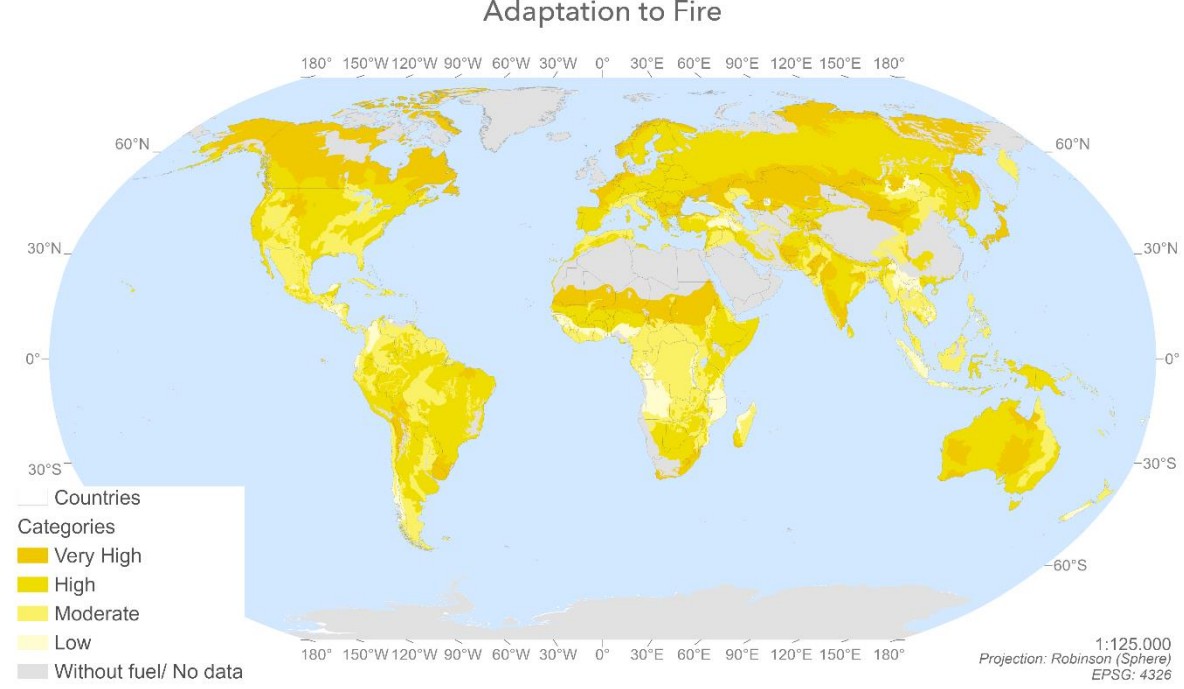

Fig. A3: Spatial distribution by ecoregion of the Ecosystem Conservation Status Value produced by combining the Indices for Unique Habitats Preservation, Intact Forest Landscapes, Degree of Fragmentation and Degree of Protection. (Source: the authors)

Fig. A4: Spatial distribution by ecoregion of the Ecosystem Adaptation to Fire Value produced by combining the Fire Regime and its degree of alteration. (Source: The authors).

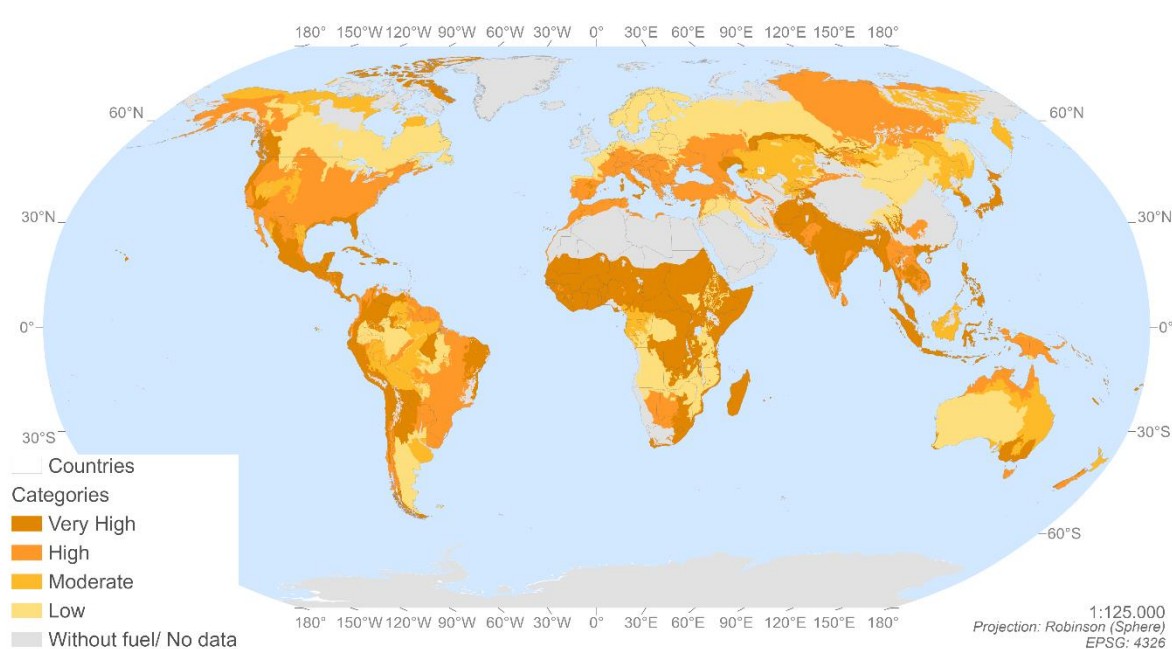

Potential Soil Erosion


**Fig. A5: Spatial distribution of Potential Soil Erosion values by ecoregion resulting from the application of**
**the FAO criterion for water erosion. (Source: The authors).**



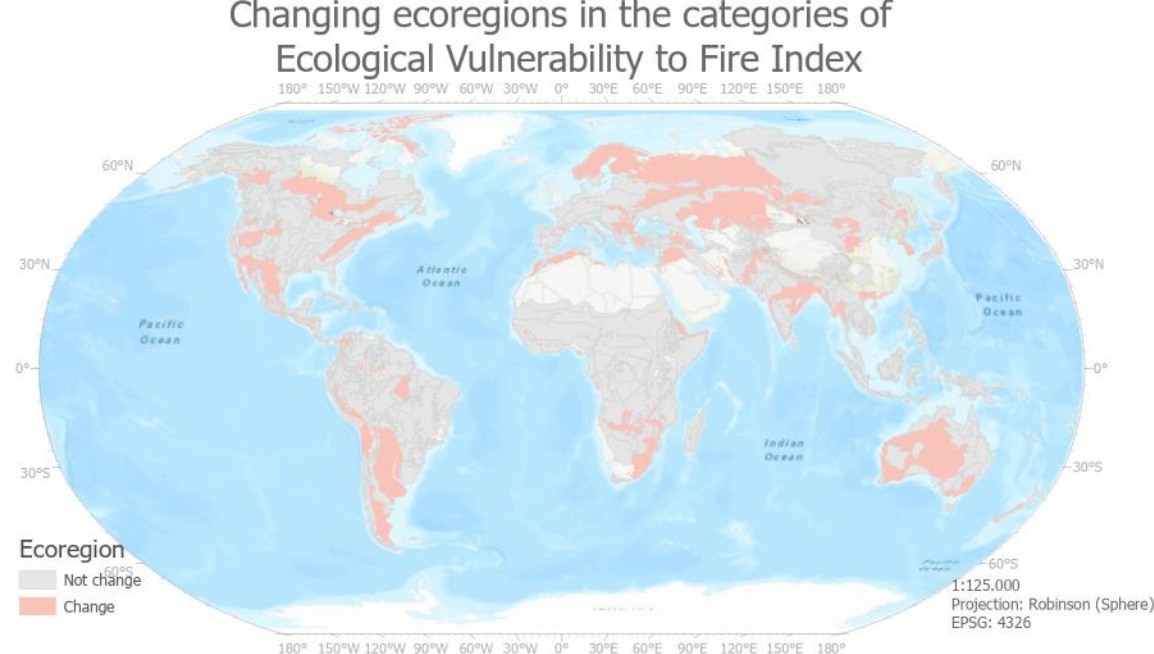

Changing ecoregions in the categories of
Ecological Vulnerability to Fire Index


**Fig. A6: Spatial distribution of changing ecoregions in the categories of Ecological Vulnerability to Fire**
**Index resulted from the OAT analyses (sensitivity method). (Source: The authors).**

**Author contributions**
Fátima Arrogante-Funes: Conceptualization, data curation, formal analysis, investigation, methodology, resources,
software, validation, visualization, writing – original draft preparation, review & editing
Inmaculada Aguado: Conceptualization, funding acquisition, investigation, methodology, project administration,
supervision, writing – review & editing.
Emilio Chuvieco: Conceptualization, funding acquisition, investigation, methodology, project administrator,
resources, supervision, writing – review & editing.
**Acknowledgements**
This research was conducted within the framework of Spanish National Project RTI2018-097538-B-I00. In
addition, Fátima Arrogante-Funes was supported by a predoctoral scholarship (FPI) from the Spanish Ministry of
Science, Innovation and Universities (PRE2019-089208). We would also like to thank ESA, Kier, the World
Wildlife Fund, Moreno-Martínez, Olson, Burgess, Potapov, Hoekstra, IUCN & UNEP-WCM, Borrelli and Shlisky
for the databases. Many thanks to the reviewers for helping to improve the manuscript.

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
