# Peer review of "Global assessment and mapping of ecological vulnerability to wildfires"

_Natural Hazards and Earth System Sciences, 2021_

## Referee Comment (RC2)

Natural Hazards and Earth System Sciences.

Review Report

Manuscript https://doi.org/10.5194/nhess-2021-285.
"Global assessment and mapping of ecological vulnerability to wildfires."

The manuscript addresses a proposal for assessing global vulnerability to wildfires using ecological value and post-fire regeneration delay indices.

It is an interesting work that is undoubtedly timely, that addresses the disturbance that fires can cause in ecosystems through pre-existing variables, which could eventually support resource management and conservation policies. The work is well supported, well structured, and well presented.

HOWEVER, I FIND A FEW MINIMUM AREAS OF OPPORTUNITY.

GLOBAL COMMENTS

The variables and indices used are the results of studies independent of this work. Thus, they are not normalized in terms of spatial resolution, which could eventually mean a problem in the integrity and uniformity of the information representation.

By combining different variables of different nature and origin, it may be possible to apply a multicriteria analysis instead of a cross-tabulation.

DETAILED COMMENTS

1. In lines 17-18, the authors mention *"The results showed that global ecological value could be reduced by as much as 50%, due to fire perturbation of ecosystems that are poorly adapted to it."* Consider rephrasing *"The results showed that global ecological value could be reduced by as much as 50% due to fire perturbation of poorly adapted ecosystems."*

2. In table 1 (line 116), "Conservation State Index," would be better to use "Status" since Status is the precision of describing the situation while State is a general description... except for your best opinion.

3. Recommendation: Specify the resolution in meters (as is done for other spatial variables) for Burnable Area (Line 134).

4. The meaning of the phrase "Monotonous linear way..." is not understood (line 195)

5. (Lines 257-259) "These variables were multiplied by their weight (Table 3) and then added together to obtain the Conservation Status Index". It is not specified how the weight values are obtained.

6. (Line 281) "two cartographies" looks like it should be "two maps." or "ecoregions cartography"

7. In the integration of the adaptation of the vegetation to fire regimes, several variables were considered (lines 282-289). The integration of the categories (fire regime and natural condition fire) explained in these same lines seems not to be objective, but subjective, which could be different if a multicriteria analysis is applied.

8. The Biome area assessment developed (lines 393-424) analyzes land cover data with the vulnerability zones in different ranges resulting from this study. It is undoubtedly important for prospecting purposes; however, it would be necessary as a context and as a contrast of the results; have carried out an analysis of the historical information on fires in a recent period with the affected land covers; which would also lead to a discussion

---

## Author Response (AR1)

**REFEREE 1**

Dear referee,

Thank you for your time on our manuscript and thoughtful comments, as well as for highlighting the weaknesses of this version. We take their recommendations very seriously and revise the manuscript accordingly. Your input is positive for us and by following your suggestions we will be able to strengthen the wording, methodology and discussion.

We hope that we have given the necessary answers to the suggestions and addressed all your doubts so that it is suitable for publication.

We have provided a detailed response to their comments below. Your comments are in bold and our responses in normal font.

**REPLY**

**GENERAL COMMENTS**

**The paper addresses an important and timely topic, with a global scale analysis. It relies on a large set of pre-existing global maps of 11 variables, or indicators, deemed relevant for the purpose of assessing ecological vulnerability to wildfires. These indicators are organized in a four-level hierarchy, with ecological vulnerability at the apex. I think the paper has problems in three domains: i) indicators/variables chosen and indices derived from them; ii) indicator/variable aggregation procedure; iii) validation and uncertainty/sensitivity analysis.**

We thank the reviewer for his-her effort in suggesting improvements in the manuscript. We will answer those concerns in the following paragraphs

**SPECIFIC COMMENTS**

**i)**

**Explanation of the meaning and justification for the choice of first-level is clear for those involved in the calculation of the second-level Biological "distinction" (BD) and Conservation "state" (CS) indices, which are well-grounded on relevant concepts from fire ecology, landscape ecology and conservation ecology. However, Potential Soil Erosion (PSE) and Adaptation to Fire (AF) indices, at the same hierarchical level, are questionable, and problems become apparent in the higher-level index resulting from their integration, Post-Fire Regeneration Delay (PFRD).**

**Regarding PSE, the authors of the RUSLE map used in the analysis (Borrelli et al., 2017) state that it does not consider the short-term effects of fire, and clarify that it deals only with land cover / land use change (Pg. 10, Methods – soil erosion modelling). The authors of the present paper need to acknowledge this and discuss how it may affect use of the RUSLE map for their assessment of ecological vulnerability. In addition, Borrelli et al. (2017) state that "…potential overall increase in global soil erosion driven by cropland expansion. The greatest**

**increases are predicted to occur in Sub-Saharan Africa, South America, and Southeast Asia…". However, Grégoire et al. (2012) and Andela et al. (2014) showed that cropland expansion in Africa, especially in the Northern Hemisphere, was responsible for substantial reductions in area burned. Taking this into account, and considering the importance of Africa in the global fire scene, does it make sense to include this indicator in the assessment of vulnerability to fire when it is associated with land use changes that strongly reduce fire incidence? An increase in erosion is expected in association with a land use change that reduces fire incidence. Similar processes may occur wherever land use intensification leads to a decrease in fire incidence. It is harder to comment on AF, because the authors are quite confused, here. They misunderstood Shlisky et al. (2007) and state that fire regimes may be "fire-dependent", "sensitive", and "independent", when these labels apply not to fire regimes, but to ecosystems, or ecoregions (more on this in Detailed comments, below). The authors need to sort out these issues.**

First, thanks for the appreciation. RUSLE has not been directly applied as a potential indicator of post-fire soil erosion. The tables that FAO considers appropriate to apply to the RUSLE have been applied in order to obtain the potential RUSLE for other phenomena FAO/UNEP/UNESCO. (1979) such as runoff, flooding, etc. Based on the bibliography, we have implemented this methodology in our study as other authors have already done (Chuvieco et al., 2014, Chuvieco et al., 2010).

On the other hand, the objective of this work is not estimating fire occurrence or burned area, but rather to estimate the potential damage that would occur in an area in case the area is burned. As it is indicated by several authors, vulnerability encompasses the potential damage that a system may suffer from different external agents (UNISDIR, 2009) (lines 54-56, 90-92). In this regard, we are interested to estimate the potential soil erosion that would be caused by the removal of forest or shrub covers when they are burned.

**PSE and AF are integrated to form PFRD, which is described as "...an indicator of the difficulties faced by the environment when recovering naturally from fire." This index is produces some starge results. How can its values be high in Zambia, NE Angola, parts of the Sudanian savannas of NH Africa, and the Llanos of Colombia/Venezuela, where Net Primary Productivity (NPP) and fire frequency are very high, but low in boreal forests, where NPP is much lower and fire return intervals much longer? What is it really measuring? Would the calculation of PFRD not benefit from incorporating NPP? In what sense can it be said that these tropical savanna areas less fire-resilient than e.g. boreal forests?**

According to different experts (Duro et al., 2007; Nagendra & Rocchini, 2008), the NPP is not a good environment proxy of post-fire vegetation recovery since high values of NPP do not imply that the vegetal cover that is settling after a fire belongs to the previous vegetal formation. The first stages after the occurrence of fires are related to herbaceous patches. And then, if there is land available to settle down between the different conditions of light and water, the woody plants would begin to proliferate. In addition, the times, the trend, the area, among others, determine the way in which this influences the NPP. If this work were to focus on a local/regional scale, the characterization of this NPP as a recovery proxy measured based on time series of fire perimeters could be possible through different techniques, as proposed by Viana-Soto et al. 2020 and Viana-Soto et al. (2022). But, for a study on a smaller work scale,

such as the global scale of this work, it would be far from representing reality since calibrating said variable for the whole world is a challenge. That is why its inclusion was ruled out for this first approximation of ecological vulnerability to fire. Likewise, in the first stages of this work, tests were carried out and the inclusion of the NPP was assessed. In addition, as reflected in the extensive bibliography, other works were reviewed in which this variable was not included in the AF part (adaptation to fire) (Turner et al., 2003; Duguy et al., 2012; Aretano et al. 2015). Subsequently, as we have explained now, we proceeded to withdraw it.

On the other hand, it should be taken into account that the indicator combines both soil erosion potential and adaptation to fire (AF), both providing a potential measure of the regeneration capacity of ecosystems or the potential response of ecosystems to fire. With this, as in other works (Chuvieco et al., 2014), the potential AF could be estimated.

On the other hand, thank you very much for providing information from different parts of the world such as Africa or South America. This will be very useful in the discussion section (new version, lines 572-576). The zones that you indicate (Africa, South America, Boreal Zones) are very specific parts of the extensive ecoregion to which they belong. It is very interesting to know in more detail these possible inconsistencies in order to explain the limitations of this model. But certainly, we consider that it is very important not to forget that the objective of this article is on a global scale and that is why the information provided is much more general than what could be expected from a local/regional scale in which the detail of those areas that you mention would have a greater differentiation (new version, lines 576-578). In addition, the input variables in some cases come from vector maps with a single data per ecoregion, as is the case of Shlisky et al., (2017). This map is generated through the transfer of knowledge from different works over the years on the ecoregion. And, based on this, hence the justification for our spatial unit to be the ecoregions. This is the information that exists on a global scale and consequently also limits the results, generalizing the different zones that make up the ecoregion.

Thank you very much for your appreciation, it has enriched the discussion of the results in the new version of the manuscript (lines 572-578).

**A second issue arises with the calculation of PFRD for the northernmost regions of Canada (High Arctic Tundra ecoregion). Why is PRFD Very High there, while the region just to the south of it (Middle Arctic Tundra) is considered "Without fuel / No data)"? How do you justify that, since it implies a reversal of the expected latitudinal gradient in vegetation abundance and fire incidence? If you had no fuel, or no data to perform the calculation in the Middle Arctic Tundra, how can you do it for the High Arctic Tundra?**

Of the 11 input variables that this study has, when in an ecoregion there is no representativeness due to the fact that some of them present for that NoData zone, information not available, incorrect information or perhaps, it does not have a vegetation cover consistent enough to able to host a fire, the ecoregion is removed. The reasons for this were, firstly, because the objective of the work is in tune with fire and that is why any ecoregion that does not contain sufficient plant cover to house said phenomenon lacks interest for the objective of this work.

Second, if we have the ecological vulnerability to fires calculated differently in each ecoregion based on different variables, the work loses the objective of being global since the ecoregions could never be comparable to each other and this would mean that they could not be estimated. vulnerability categories for the whole world, for this reason the ecoregions that cannot host these 11 starting variables were left out.

**ii)**

**The authors aggregate their variables/indicators into first-level indices, and then aggregate these indices up the hierarchy using cross-tabulation. This is approach, which is simple to implement, has drawbacks. It requires variable discretization, which wastes information, and requires hard to justify, implicit decisions on the numbers of classes, and on the positions of the thresholds between classes. It ignores issues of compensation between indicators and implicitly weights all indicators equally. At a minimum, the authors need to justify these implicit decisions, but it would be preferable to aggregate the variables/indicators using one of several available multicriteria methods, namely those revised by El Gibari et al. (2019) for the specific purpose of building composite indicators, or indices.**

Thanks for the suggestions. The multicriteria analysis for a local/regional scale is interesting and enriching since it is easier to find a panel of experts representative of the territory. But in contrast, for a global scale it would not be relevant given the difficulty of global experts. This would result in a biased study based on the territories of which it was or was not representative (Borrero & Henao, 2017; Hämäläinen & Alaja, 2008). For this reason, it was decided to use the cross-tabulation integration methodology that tries to be as objective as possible, also used in spatial studies at global scale (Chuvieco et al., 2014) or at local/regional scale (Arrogante-Funes et al., 2020; Martínez-Vega et al., 2007). The justification of that it has done in the new version of the manuscript in the introduction section (lines 65-71), methodology (lines 104-106) and discussion (lines 579-580).

Based on this first study and initiation on the global scale of ecological vulnerability to fires, we have detected the limitations of using classic heuristic methods and that is why we are developing improvements using Auto ML models and Fuzzy algorithms in order to avoid bias. that these methods cause (Bruzón et al. 2021). Certainly, these new works arise from this first exploration and that is why implementing something different would be a new work and would not fulfill its idea.

In relation to justifying the intervals, given the disparity of the sample due to having such an extremely large study area, it was decided to divide it according to quantiles. This decision will be justified based on bibliography in which this method is used, through works such as Pereira et al., (2020), Xing & Ree, (2017), among others (new version, lines 247-248).

**iii)**

**Validation of composite indices often is problematic, because they deal with unmeasurable criteria, or are not meant to predict an effective impact but to estimate a risk or a potential effect (Bockstaller and Girardin, 2003, Moriarty et al., 2018). This is the case for the present**

**paper, and the authors acknowledge it in lines 558-560. However, that does not imply the issue can be ignored, or postponed for future research, as the authors propose to do. Given the constraints on empirical validation of the proposed index, it becomes especially important to focus on conceptual, or design validation (Bockstaller and Girardin, 2003), i.e. assessment of the scientific quality of the construction or design of the index, and on sensitivity / uncertainty analysis of the implications of decisions made while constructing the index (Saisana et al., 2005; Tate, 2012), Therefore, I urge the authors to strengthen their defense of the scientific quality of the index, both in terms of the variables chosen and the way they are aggregated. They also need to perform a sensitivity analysis of the key implications of the decisions implicit in variable discretization and in the chain of cross-tabulations implemented. This is essential to demonstrate the validity and reliability of the index and to facilitate its proper use.**

We have carried out a sensitivity analysis on the integration in the index of ecological vulnerability to fires called One-at-a-time (Saltelli et al., 2000). Within the bibliographic review (Gonzalez et al., 2015; Clavijo et al., 2019), it is one of the methods that admits, modifying it, variations to be able to compute with categorical variables as in our case. When working with categorical variables, the use of the described sensitivity methods that are based on formulas and numerical algorithms becomes unfeasible for the case of categorical variables (Saltelli et al., 2000). Through the OAT, it is possible to have an idea of which ecoregions have a consistent value of the ecological vulnerability index to fires, as opposed to which ones present greater uncertainty, as well as a percentage of coincidences of the model itself. Despite this, as the literature points out, a qualitative integration method is neither better nor worse than a quantitative one, since they all have their uncertainty with which they must deal (Richards & Rowe, 1999; Goodchild et al., 1993; Heuvelink, 1998; Heuvelink et al., 1989).

This changes have been done in the new version of the manuscript (lines 323-341, 450-463, 568-580, 912-915).

On the other hand, all the variables used as well as their processing and their integration in the different indicators and index are fully justified based on the literature, as can be seen throughout the text (lines 108-110, 123-125, 147- 149,178-180, 182-184, 186-187, 199-203, 211-213, 216-220, 229-231, 236-240, 241-243, 247-248, 254-256, 262-264, 268- 274, 279-283, 288-291, 294-297, 304-310, 318-319).

**References**

**Borrelli, P., Robinson, D. A., Fleischer, L. R., Lugato, E., Ballabio, C., Alewell, C., ... & Panagos, P. (2017). An assessment of the global impact of 21st century land use change on soil erosion. Nature Communications, 8(1), 1-13.**

**Grégoire, J. M., Eva, H. D., Belward, A. S., Palumbo, I., Simonetti, D., & Brink, A. (2012).**

**Effect of land-cover change on Africa's burnt area. International Journal of Wildland Fire, 22(2), 107-120.**

**Andela, N., & Van Der Werf, G. R. (2014). Recent trends in African fires driven by cropland expansion and El Niño to La Niña transition. Nature Climate Change, 4(9), 791-795.**

El Gibari, S., Gómez, T., & Ruiz, F. (2019). Building composite indicators using multicriteria methods: a review. Journal of Business Economics, 89(1), 1-24.

Bockstaller, C., & Girardin, P. (2003). How to validate environmental indicators.

Agricultural Systems, 76(2), 639-653.

Moriarty, P. E., Hodgson, E. E., Froehlich, H. E., Hennessey, S. M., Marshall, K. N., Oken, K. L., ... & Stawitz, C. C. (2018). The need for validation of ecological indices. Ecological Indicators, 84, 546-552.

Saisana, M., Saltelli, A., & Tarantola, S. (2005). Uncertainty and sensitivity analysis techniques as tools for the quality assessment of composite indicators. Journal of the Royal Statistical Society: Series A (Statistics in Society), 168(2), 307-323.

Tate, E. (2012). Social vulnerability indices: a comparative assessment using uncertainty and sensitivity analysis. Natural Hazards, 63(2), 325-347.

**CORRECTIONS (AND A FEW MORE SPECIFIC COMMENTS)**

**Line 13: "biological distinction" does not sound right in English. I believe "biological distinctiveness" is preferable. Please correct throughout the text.**

It has been done throughout the text.

**Lines 15-16: why did you choose to combine the various indicators using qualitative crosstabulation? This option needs a justification, because there are alternatives, e.g. multicriteria evaluation.**

Thank for the appreciation. It has done throughout the text (lines 65-70, 104-106, 579-580).

Based on the various previous studies that used this method, we are going to justify the use of qualitative cross tabulation with it, such as the works by Arrogante-Funes et al., (2020), Chuvieco et al., (2014), Martínez-Vega et al., (2007), Isabel et al., (2003), among others.

On the other hand, the multicriteria analysis for a local/regional scale is interesting and enriching since it is easier to find a panel of experts representative of the territory. But in contrast, for a global scale it would not be relevant given the difficulty of finding representatives of the entire territory of the Earth. This would result in a biased study based on the territories of which it was or was not representative (Borrero & Henao, 2017; Hämäläinen & Alaja, 2008). For this reason, it was decided to use the cross-tabulation integration methodology that tries to be as objective as possible, also used in the previous spatial studies (previous paragraph).

**Line 37: forest "masses" is not used in English. Please replace with "stands", or "patches".**

Thanks for the appreciation. It has been done.

**Line 41: Are you really talking about fires in forests, only? Or are you using the term in a broader (and inappropriate) sense of vegetation fires?**

In any case, and following your recommendation, the document has been rigorously revised. In the document, as can be seen in its title, the term Wildlfire appears, it has been proposed to replace by Forest Fire.

**Lines 47-50: In terms of natural hazards terminology, I recommend using the United Nations International Strategy for Disaster Reduction (UNISDR) terminology on disaster risk reduction (2009). It considers that risk assessment involves the combination of hazard, exposure, and vulnerability, according to the definitions proposed in that glossary.**

Thanks for the recommendation. It has been done.

**Line 62-64: Exposure is not a part of vulnerability and is defined somewhat differently from the way you use it. I quote from UNISDR, 2009: "Exposure - People, property, systems, or other elements present in hazard zones that are thereby subject to potential losses. Comment: Measures of exposure can include the number of people or types of assets in an area. These can be combined with the specific vulnerability of the exposed elements to any particular hazard to estimate the quantitative risks associated with that hazard in the area of interest."**

Thanks for the suggestion and for the clarification. It has been done.

**Line 63: I believe that the standard use in the specialized literature is to define "index" as the result of aggregating two or more "indicators". This terminology has the advantage of distinguishing different levels of the analysis hierarchy. You use "index" for all levels, which occasionaly is confusing. Please consider adopting the distinction indicator/index in the text.**

Thank you for the recommendation and the effort to make the manuscript as clear as possible. We have rigorously reviewed the text and we have changed all of it. For Ecological index, we have chosen Ecological Indicator and for Post-Fire Regeneration Delay index, que have chosen indicator too. For the Ecological Vulnerability to Fire we have maintained index, as you propose.

**Lines 80-81: This explanation of how adaptation to fire was estimated is too vague, please elaborate, including identification of the dynamic global vegetation model used in the analysis. Otherwise, it is not possible to evaluate the adequacy of this indicator.**

In the introduction, in order to simplify the extensive bibliographic review, it was decided to summarize with what you point out. The digital vegetation model used was ORCHIDEE and it has been mentioned in the new version of the manuscript.

**Line 85: is "exceptionality" what was previously called "distinction"? Please clarify and use consistent terminology, avoiding "distinction".**

The term exceptionality refers to the Representativeness Criteria which described in lines 139-147 of this manuscript: "…In this way, each biome is guaranteed to have at least one priority ecoregion, so ensuring, for example, that the ecoregions in the savannah forest biome can also be classified, in addition to the more important moist tropical forests, which would otherwise dominate the list of values due to their high rates of species richness and endemic species…".

Distinction will be replaced by distinctiveness. So, distinctiveness is related to Biological Distinctiveness Index which described in lines 153-211 of this manuscript: "… is more than just biodiversity at the species level, in that it also covers the diversity of ecological functions and the processes that support structural biodiversity (Ricketts et al., 1999). Specifically, this study is based on taxonomic rarity, species richness, functional diversity, and habitats with a unique evolution…".

**Lines 114-116: Terminology: "distinction", "index". Also "state", where it should read "status".**

Thank you for the recommendation and the effort to make the manuscript as clear as possible. We have rigorously reviewed the text and we have changed all of it.

**Line 127: The value 14 did not change.**

Perhaps, a correct rephrasing should be: "In this way, the final number of ecoregions was 660, having representation of all terrestrial biomes."

It has been done in lines 120-121.

**Line 136-137: Already mentioned, can be deleted.**

It has been done.

**Line 195: "Monotonic", not "monotonous".**

It has been done throughout the text.

**Line 258: Where do the weights come from, how were they obtained? Table 3 shows "Maximum scores", not weights. Are they the same thing? If so, please consistently use a single term to refer to the concept.**

Yes, they are the same things. It has been done.

**Line 267: You don't integrate the index, you integrate lower level indicators, to create the index.**

It has been done throughout the text.

**Line 280: This section is confusing. It is not fire regimes that are "fire-dependent", "sensitive", and "independent", it is the structure and function of ecosystems, or ecoregions. Notice that a fire-independent fire regime would be a nonsensical concept. See Shlisky et al. (2007), pg. 5: "Ecosystems can be classified in terms of their relationship to fire regime characteristics such as fuels, flammability, ignitions, and fire spread conditions within a given ecosystem." The authors need to improve their understanding of the concepts they are using here.**

Thanks for finding the weakness of the manuscript. We have recalled "Relationship between fire and ecoregion" throughout the text.

**Line 281: "Maps", not "cartographies".**

It has been done.

**Line 314: Is a "factor" the same as an "indicator"? Please use the technical terminology consistently.**

It has been corrected throughout the text.

**Line 330: I don't understand the meaning of "in which the most valuable component was prioritized". Please clarify. The procedure described here is a cross-tabulation of a crosstabulation. It is pertinent to question the sensitivity of your results to the decisions implicit in the methodology, namely number of levels and placement of thresholds, especially when accumulating the results of successive cross-tabulations. I realize that to the paper is essentially normative, in the sense that it prescribes a procedure to assess an index that is not directly measurable, and for which empirical validation may not be feasible, as the authors acknowledge. However, this should not exempt the authors from performing a sensitivity and uncertainty analysis of the implications of decisions on the discretization of quantitative variables, and on the procedures used to weight and integrate them.**

The phrase **"in which the most valuable component was prioritized"** means that when two categories coincide, they are incremented to the higher level.

All levels and the different categories of the variables are in the main document in the tables:

- Biological distinctiveness: Lines 196-197
- Conservation Status: Lines 249-252
- Ecological Indicator: Lines 258-260
- Adaptation to the Vegetation to Fire: Lines 285-286
- Soil Erosion Potential: Lines 299-300
- Post Fire Vegetation Regeneration Delay Indicator: Lines 312-313
- Ecological Vulnerability to Wildfire Index: Lines 321-322

Regarding the sensitivity analysis, OAT has done (Saltelli et al., 2000), as we have said previously (lines 323-341, 450-463, 568-580, 912-915).

**Lines 346-347: Florida and Thailand are not located in temperate zones of the globe.**

**Line 363: …neither is the Yucatan Peninsula.**

Thanks for the clarification of both cases. It has been deleted from it.

**Line 366: It is surprising to see Zambia and NE Angola mapped with a very high Post-fire Regeneration Delay, especially considering how often they burn. Please clarify this apparent inconsistency.**

The zones that you indicate (Africa, South America, Boreal Zones) are very specific parts of the extensive ecoregion to which they belong. It is very interesting to know in more detail these possible inconsistencies in order to explain the limitations of this model. But certainly, we consider that it is very important not to forget that the objective of this article is on a global scale and that is why the information provided is much more general than what could be expected from a local/regional scale in which the detail of those areas that you mention would have a greater differentiation. In addition, the input variables in some cases come from vector maps with a single data per ecoregion, as is the case of Shlisky et al., (2017) or the tables of biodiversity provided by World Wildlife Fund, (2006), among others.

**Line 375: Are "potential ecological damages" what is called Ecological Value before and after this point? Please clarify and use the terminology consistently.**

Yes, they are the same. It has been done.

**Line 393: Section 3.3.2.: This section reports too many numbers in text format. Just stress the key points of Table 10 in the text and use charts and graphs to summarize the rest, if necessary.**

We chose a table to show our results because a number is an exactly result whereas the graphic in which the reader has to find an approximated value. We find this visualization of the result more interesting for the police makers, among others.

**Line 395: Are "Ecological Indices" yet another name to "Ecological Value"? Please clarify and keep the terminology consistent.**

Again, sorry for the confusing of the different terminology. And yes, they are the same. It has been done throughout the text.

**Line 396: Your analysis is static. How can it be influenced by fire trends? What do you mean? Also, several of the vulnerable areas are not Forests, e.g. Tundra and Mangroves. Are you using "forest fire" to refer to all vegetation fires, regardless of the type of ecosystem where they occur? Please avoiding doing this and use the more generic expression "vegetation fires" when not strictly referring to fires in forests. The same applies to the text in lines 401-403.**

As indicated earlier the term forest fire has been replaced by Wildfires to avoid potential confusions as indicated by the reviewer. as written in the title of the manuscript.

**Line 557: "intuitive" does not sound like a good term, since it is the opposite of objective and rational. I suggest you replace the term by "easily understood", or something similar.**

Thanks for your appreciation. It has been done.

**Lines 579-580: The unfeasibility of empirical validation of your index against a set of objective, measurable data makes it especially important to perform a sensitivity / uncertainty analysis, so that potential users understand the strengths and weaknesses of the index, and use it properly. It cannot be postponed to a subsequent paper and should be included in this one.**

As indicated earlier, the described sensitivity methods that are based on formulas and numerical algorithms becomes unfeasible for the case of categorical variables (Saltelli et al., 2000). Through the OAT, it is possible to have an idea of which ecoregions have a consistent value of the ecological vulnerability index to fires, as opposed to which ones present greater uncertainty, as well as a percentage of coincidences of the model itself. Despite this, as the literature points out, a qualitative integration method is neither better nor worse than a quantitative one, since they all have their uncertainty with which they must deal (Richards & Rowe, 1999; Goodchild et al., 1993; Heuvelink, 1998; Heuvelink et al., 1989).

This changes have been done in the new version of the manuscript (lines 323-341, 450-463, 568-580, 912-915).

Clavijo, A. U., Delgado, M. G., & González, P. B. (2019). Análisis de Sensibilidad aplicado a modelos de crecimiento urbano basados en autómatas celulares de estructura irregular. Cuadernos Geográficos, 58, 326–348.

Gonzalez, P. B., Aguilera-Benavente, F., & Gomez-Delgado, M. (2015). Partial validation of cellular automata based model simulations of urban growth: An approach to assessing factor influence using spatial methods. Environmental Modelling & Software, 69. https://doi.org/10.1016/j.envsoft.2015.03.008

Goodchild, M. F., Parks, B. O., & Steyaert, L. T. (1993). Environmental Modelling with GIS. Environmental Modelling with GIS. Oxford University Press, New York, 318–331.

Heuvelink, G. B. (1998). Error propagation in environmental modelling with GIS. CRC Press.

Heuvelink, G. B., Burrough, P. A., & Stein, A. (1989). Propagation of errors in spatial modelling with GIS. International Journal of Geographical Information System, 3(4), 303–322.

Richards, D., & Rowe, W. D. (1999). Decision-making with heterogeneous sources of information. Risk Analysis, 19(1), 69–81.

Saltelli, A., Chan, K., & Scott, E. . (2000). Sensitivity Analysis. Chichester, John Wiley & Sons, LTD.

Chuvieco, E., Aguado, I., Yebra, M., Nieto, H., Salas, J., Martín, M. P., Vilar, L., Martínez, J., Martín, S., Ibarra, P., de la Riva, J., Baeza, J., Rodríguez, F., Molina, J. R., Herrera, M. A., & Zamora, R. (2010). Development of a framework for fire risk assessment using remote sensing and geographic information system technologies. Ecological Modelling, 221(1), 46–58. https://doi.org/10.1016/j.ecolmodel.2008.11.017

Chuvieco, E., Martínez, S., Román, M. V., Hantson, S., & Pettinari, M. L. (2014). Integration of ecological and socio-economic factors to assess global vulnerability to wildfire. Global Ecology and Biogeography, 23(2), 245– 258. https://doi.org/10.1111/geb.12095

FAO/UNEP/UNESCO. (1979). A provisional methodology for soil degradation assessment. Food and 692 Agricultural Organization of the United Nations, Rome.

Mildrexler, D. J., Zhao, M., & Running, S. W. (2009). Testing a MODIS global disturbance index across North America. Remote Sensing of Environment, 113(10), 2103-2117.

Borrero, S., & Henao, F. (2017). Can managers be really objective? Bias in multicriteria decision analysis. Academy of Strategic Management Journal, 16(1), 244-259.

Hämäläinen, R. P., & Alaja, S. (2008). The threat of weighting biases in environmental decision analysis. Ecological Economics, 68(1-2), 556-569.

Isabel, M. P. M., Calcerrada, R. R., & Vega, J. M. (2003). Valoración del paisaje en la zona de especial protección de aves carrizales y sotos de Aranjuez (Comunidad de Madrid). Geofocus: Revista Internacional de Ciencia y Tecnología de la Información Geográfica, (3), 2.

Martínez Vega, J., Romero Calcerrada, R., & Echavarría Daspet, P. (2007). Valoración paisajística y ecológica de la Comunidad de Madrid: su integración en un índice sintético de riesgo de incendios forestales.

Shlisky, A., Waugh, J., Gonzalez, P., Gonzalez, M., Manta, M., Santoso, H., Alvarado, E., Ainuddin, A., Rodríguez-trejo, D. A., Swaty, R., Schmidt, D., Kaufmann, M., Myers, R., Alencar, A., Kearns, F., Johnson, D., Smith, J., & Zollner, D. (2007). Fire, ecosystems and people: threats and strategies for global biodiversity conservation. The Nature Conservancy Global Fire Initiative Technical Report, 17. http://mrcc.isws.illinois.edu/living_wx/wildfires/fire_ecosystems_and_people.pdf

World Wildlife Fund. (2006). WildFinder: Online database of species distributions.

Arrogante-Funes, P., Bruzón, A. G., Arrogante-Funes, F., Ramos-Bernal, R. N., & Vázquez-Jiménez, R. (2021). Integration of vulnerability and hazard factors for landslide risk assessment. International Journal of Environmental Research and Public Health, 18(22). https://doi.org/10.3390/ijerph182211987

Bruzón, A. G., Arrogante-Funes, P., Arrogante-Funes, F., Martín-González, F., Novillo, C. J., Fernández, R. R., ... & Ramos-Bernal, R. N. (2021). Landslide susceptibility assessment using an AutoML framework. International journal of environmental research and public health, 18(20), 10971.

Pereira, H. M., Rosa, I. M., Martins, I. S., Kim, H., Leadley, P., Popp, A., ... & Alkemade, R. (2020). Global trends in biodiversity and ecosystem services from 1900 to 2050. bioRxiv (Preprint).

Xing, Y., & Ree, R. H. (2017). Uplift-driven diversification in the Hengduan Mountains, a temperate biodiversity hotspot. Proceedings of the National Academy of Sciences, 114(17), E3444-E3451.

Aretano, R., Semeraro, T., Petrosillo, I., De Marco, A., Pasimeni, M. R., & Zurlini, G. (2015). Mapping ecological vulnerability to fire for effective conservation management of natural protected areas. Ecological Modelling, 295, 163–175. https://doi.org/10.1016/j.ecolmodel.2014.09.017

Duguy, B., Alloza, J. A., Baeza, M. J., De La Riva, J., Echeverría, M., Ibarra, P., Llovet, J., Cabello, F. P., Rovira, 680 P., & Vallejo, R. V. (2012). Modelling the ecological vulnerability to forest fires in mediterranean ecosystems 681 using geographic information technologies. Environmental Management, 50(6), 1012–1026. 682 https://doi.org/10.1007/s00267-012-9933-3

Duro, D. C., Coops, N. C., Wulder, M. A., & Han, T. (2007). Development of a large area biodiversity monitoring system driven by remote sensing. Progress in Physical Geography, 31(3), 235–260. https://doi.org/10.1177/0309133307079054

Nagendra, H., & Rocchini, D. (2008). High resolution satellite imagery for tropical biodiversity studies: The devil is in the detail. Biodiversity and Conservation, 17(14), 3431–3442. https://doi.org/10.1007/s10531-008-9479-0

Turner, B. L., Kasperson, R. E., Matsone, P. A., McCarthy, J. J., Corell, R. W., Christensene, L., Eckley, N., Kasperson, J. X., Luers, A., Martello, M. L., Polsky, C., Pulsipher, A., & Schiller, A. (2003). A framework for vulnerability analysis in sustainability science. Proceedings of the National Academy of Sciences of the United States of America, 100(14), 8074–8079. https://doi.org/10.1073/pnas.1231335100

Viana-Soto, A., Aguado, I., Salas, J., & García, M. (2020). Identifying post-fire recovery trajectories and driving factors using landsat time series in fire-prone mediterranean pine forests. Remote Sensing, 12(9), 1499.

Viana-Soto, A., García, M., Aguado, I., & Salas, J. (2022). Assessing post-fire forest structure recovery by combining LiDAR data and Landsat time series in Mediterranean pine forests. International Journal of Applied Earth Observation and Geoinformation, 108, 102754.

**REFEREE 2**

Dear referee,

Thank you for your time on our manuscript and thoughtful comments, as well as for highlighting the weaknesses of this version. We take their recommendations very seriously and revise the manuscript accordingly.

Your input is positive and by following your suggestions we will be able to strengthen the wording and understanding will need more detail as needed as well as justify the selection of integration approach for the global scale.

We hope that we have given the necessary answers to the suggestions and addressed all your doubts so that it is suitable for publication.

We have provided a detailed response to their comments below. Your comments are in bold and our responses in normal font.

**REPLY**

**The manuscript addresses a proposal for assessing global vulnerability to wildfires using ecological value and post-fire regeneration delay indices.**

**It is an interesting work that is undoubtedly timely, that addresses the disturbance that fires can cause in ecosystems through pre-existing variables, which could eventually support resource management and conservation policies. The work is well supported, well structured, and well presented.**

First of all, we would like to thank the reviewer for his/her review of the manuscript, constructive comments and suggestions to improve it. We have carefully considered his/her comments, especially to clarity the writing of the manuscript as well as the methodology used to cross the variables.

**Global**

**The variables and indices used are the results of studies independent of this work. Thus, they are not normalized in terms of spatial resolution, which could eventually mean a problem in the integrity and uniformity of the information representation.**

We were aware of this challenge. Considering the target global scale of our research, we decided to select the ecoregion as the basic spatial unit. We computed values at this geographical unit through different statistical methods as indicated in the bibliography. In any case, we have account for the areas of each ecoregion to standardize the different input databases.

**Detailed Comments**

**In lines 17-18, the authors mention "The results showed that global ecological value could be reduced by as much as 50%, due to fire perturbation of ecosystems that are poorly adapted to it." Consider rephrasing "The results showed that global ecological value could be reduced by as much as 50% due to fire perturbation of poorly adapted ecosystems."**

Thanks, it has been done.

**In table 1 (line 116), "Conservation State Index," would be better to use "Status" since Status is the precision of describing the situation while State is a general description... except for your best opinion.**

It has done throughout the text, sorry for the confusion since what you explain is exactly what we wanted to express. Thanks for the clarification.

**Recommendation: Specify the resolution in meters (as is done for other spatial variables) for Burnable Area (Line 134).**

The information suggested is in the manuscript on different lines due to the previous process applied. The information was in line 126: "... 300m". Then, the database was resampled at 0.25 deg (line 134) which is the resolution of our product.

**The meaning of the phrase "Monotonous linear way..." is not understood (line 195)**

We meant that all values of the different variables take the range from 1 to 100 through a linear function: "First, the factors per ecoregion were scaled in a monotonic linear way taking the range from 1 to 100 through a linear function within the biome (particular for each variable, depending on the maximum and minimum value of it)."

**(Lines 257-259) "These variables were multiplied by their weight (Table 3) and then added together to obtain the Conservation Status Index". It is not specified how the weight values are obtained.**

The information suggested is in the manuscript and this comes from the cited bibliography, line 257: "...as proposed by Burgess et al., (2006) and by Ricketts et al., (1999)."

Maximum scores have changed by weights in order to improve the explanation of it (lines 242-246, 252 (table 3)).

**(Line 281) "two cartographies" looks like it should be "two maps." or "ecoregions cartography"**

It has done.

**In the integration of the adaptation of the vegetation to fire regimes, several variables were considered (lines 282-289). The integration of the categories (fire regime and natural condition fire) explained in these same lines seems not to be objective, but subjective, which could be different if a multicriteria analysis is applied.**

Thanks for the suggestion of the multicriteria analysis tool. Thanks for the suggestions. The multicriteria analysis for a local/regional scale is interesting and enriching since it is easier to find a panel of experts representative of the territory. But in contrast, for a global scale it would not be relevant given the difficulty of finding global wildfire experts. This would result in a biased study based on the territories of which it was or was not representative (Borrero & Henao, 2017; Hämäläinen & Alaja, 2008). For this reason, it was decided to use the cross-tabulation integration methodology that tries to be as objective as possible, also used in spatial studies at global scale (Chuvieco et al., 2014) or at local/regional scale (Arrogante-Funes et al., 2020; Martínez-Vega et al., 2007). The justification of that it has done in the new version of the manuscript in the introduction section (lines 65-71), methodology (lines 104-106) and discussion (lines 579-580).

Regarding subjectivity, we have included a sensitivity analysis, as suggested by the first reviewer, to mitigate the possible bias of subjectivity. In addition, the crossings are based on suggestions from the authors of the different variables.

Based on this first study and initiation on the global scale of ecological vulnerability to fires, we have detected the limitations of using classic heuristic methods and that is why we are developing improvements using Auto ML models and Fuzzy algorithms in order to avoid bias. that these methods cause (Bruzón et al. 2021). Certainly, these new works arise from this first exploration and that is why implementing something different would be a new work and would not fulfill its idea.

In relation to the intervals, given the disparity of the sample due to having such an extremely large study area, it was decided to divide it according to quantiles. This decision will be justified based on bibliography in which this method is used, through works such as Pereira et al., (2020), Xing & Ree, (2017), among others (new version, lines 247-248).

**The Biome area assessment developed (lines 393-424) analyzes land cover data with the vulnerability zones in different ranges resulting from this study. It is undoubtedly important for prospecting purposes; however, it would be necessary as a context and as a contrast of the results; have carried out an analysis of the historical information on fires in a recent period with the affected land covers; which would also lead to a discussion.**

One of the objectives of this work is to obtain the vulnerability to fires on a global scale and the databases and methodology used in it take into account a wide period of time in order to collect the average behavior or pattern of the variables. With this, a vision "of the moment" of said vulnerability is achieved. On the other hand, as some experts on this subject (Adger, 2006; Birkman, 2006, Kienberg, 2013) pointed out, there are different spatial-temporal scales to carry out this study and each of them is equally accepted depending on the initial objective. In addition, we wanted to focus on spatial variation and that in the future we plan to cover

temporal variation, but this is problematic because there are no sufficiently long and high-quality global fire databases, except for the MODIS series from 2001 to 2021.

For this reason, the suggestion of taking into account the most recent years would be of interest in the analysis of vulnerability presenting different scenarios. This is precisely the continuation of our line of research. Again, thank you for the interest shown.

Adger, W. N. (2006). Vulnerability. Global environmental change, 16(3), 268-281.

Birkman, J. 2006. Measuring Vulnerability to Natural Hazards: Towards Disaster Resilient Societies.

Kienberger, S., Blaschke, T., & Zaidi, R. Z. (2013). A framework for spatio-temporal scales and concepts from different disciplines: the 'vulnerability cube'. Natural Hazards, 68(3), 1343-1369.

Bruzón, A. G., Arrogante-Funes, P., Arrogante-Funes, F., Martín-González, F., Novillo, C. J., Fernández, R. R., ... & Ramos-Bernal, R. N. (2021). Landslide susceptibility assessment using an AutoML framework. International journal of environmental research and public health, 18(20), 10971.

Hämäläinen, R. P., & Alaja, S. (2008). The threat of weighting biases in environmental decision analysis. Ecological Economics, 68(1–2), 556–569. https://doi.org/10.1016/j.ecolecon.2008.05.025

Martínez Vega, J., Romero Calcerrada, R., & Echavarría, P. (2007). Valoración paisajística y ecológica de la Comunidad de Madrid: su integración en un índice sintético de riesgo de incendios forestales. Revista de Teledetección, 28(April 2016), 43–60.

Pereira, H. M., Rosa, I. M., Martins, I. S., Kim, H., Leadley, P., Popp, A., ... & Alkemade, R. (2020). Global trends in biodiversity and ecosystem services from 1900 to 2050. bioRxiv (Preprint).

Xing, Y., & Ree, R. H. (2017). Uplift-driven diversification in the Hengduan Mountains, a temperate biodiversity hotspot. Proceedings of the National Academy of Sciences, 114(17), E3444-E3451.

Arrogante-Funes, P., Bruzón, A. G., Arrogante-Funes, F., Ramos-Bernal, R. N., & Vázquez-Jiménez, R. (2021). Integration of vulnerability and hazard factors for landslide risk assessment. International Journal of Environmental Research and Public Health, 18(22). https://doi.org/10.3390/ijerph182211987

Borrero, S., & Henao, F. (2017). Can managers be really objective? Bias in multicriteria decision analysis. Academy of Strategic Management Journal, 16(1), 244–260. https://www.abacademies.org/articles/can-managers-be-really-objective-Bias-in-multicriteria-decision-analysis-1939-6104-16-1-114.pdf

Chuvieco, E., Martínez, S., Román, M. V., Hantson, S., & Pettinari, M. L. (2014). Integration of ecological and socio-economic factors to assess global vulnerability to wildfire. Global Ecology and Biogeography, 23(2), 245– 258. https://doi.org/10.1111/geb.12095

---

## Author Response (AR2)

**Response Referee 1**

**Line 80: ORCHIDEE model. Please provide reference.**

Done.

Lines 80-81 and 846-848: Kinner et al., (2005)

**Line 195: "(particular for each variable, depending on the maximum and minimum value of it)." This continues to sound weird. I suggest you check with a native English speaker to fix it. I don't understand what you mean.**

Done.

Lines 193-194: First, the factors were scaled between 1 and 100 through a linear function per biome.

**Line 261: Pereira et al. (2020) is missing from the list of references.**

Done.

Line 260 and 895: Pereira, H. M., Rosa, I. M. D., Martins, I. S., & Al., E. (2020). Supplementary Materials for Global trends in biodiversity and ecosystem services from 1900 to 2050. Science, 1, 1–5.

**Line 282: After "Firstly…" please clarify who performed the grouping reported: yourselves, or Shlisky et al. (2007)?**

Done.

Line 281: Firstly, in this database,…

**Line 284: Are you specifically talking about forests, or are you using the expression "forest fire" to mean the broader concept of "vegetation fire"?**

Done.

Line 283: management tool (deleted forest)

**Line 286: The misunderstanding I pointed out earlier remains uncorrected here: a fire-independent fire regime is a contradiction in terms. It's ecoregions that can be fire-independent, not fire regimes. Without fire, there is no fire regime.**

Done.

Lines 285-286: The first grouping includes fire-dependent, sensitive and independent ecoregions, while the second classifies ecoregions according to intact, altered and highly altered respect the first classification.

**Line 338: please correct "on-at-a-time", it's "one-at-a-time".**

Done.

Line 337: One-at-a-time

**Line 388: "the most resilient areas of the planet (very high or high Adaptation to Fire values and low or moderate Potential Soil Erosion) are in the temperate broadleaf and mixed forests of northern Europe…". Fire a uncommon event in these forests. Please justify these scores in ecosystems with very limited evolutionary exposure to fire.**

Done.

Line 388: are in the temperate broadleaf and mixed forests of the northern Europe (deleted). Line 588-592: Another example would be that the most resilient areas on the planet (very high or high Fire Adaptation values and low or moderate Potential Soil Erosion) are found in the temperate broadleaf and mixed forests of northern Europe when fire is a rare event in these ecoregions and thus lack a history of fire-attuned evolution.

**Line 471: Punctuation: "High, (insert comma here) reaching higher numbers of ecoregions (+95) and Low, (insert comma here) decreasing its…".**

Done.

Line 570: High, reaching higher numbers of ecoregions (+95) and Low, decreasing its number of ecoregions considerably to 14 (-65).

**Line 583-4: The issue described here is a flaw of the method, which should be acknowledged as such. Please mention that it will be corrected in future revisions of this work. This is the type of bias that may affect the formulation of public policies, not to mention that it may discredit the work in the eyes of fire ecologists from these ecoregions.**

Done.

Line 597: Despite this, these uncertainties will be explored for future versions.